# Systemic hypoxia inhibits T cell response by limiting mitobiogenesis via matrix substrate-level phosphorylation arrest

Amijai Saragovi[1], Ifat Abramovich[2], Ibrahim Omar[1], Eliran Arbib[1], Ori Toker[3], Eyal Gottlieb[2], Michael Berger[1]*

[1]The Lautenberg center for Immunology and Cancer Research, The Institute for Medical Research Israel-Canada, The Hebrew University Medical School, Jerusalem, Israel; [2]The Ruth and Bruce Rappaport, Faculty of Medicine, Technion - Israel Institute of Technology, Jerusalem, Israel; [3]Faculty of Medicine, Hebrew University of Jerusalem; The Allergy and Immunology Unit, Shaare Zedek Medical Center, Jerusalem, Israel

**Abstract** Systemic oxygen restriction (SOR) is prevalent in numerous clinical conditions, including chronic obstructive pulmonary disease (COPD), and is associated with increased susceptibility to viral infections. However, the influence of SOR on T cell immunity remains uncharacterized. Here we show the detrimental effect of hypoxia on mitochondrial-biogenesis in activated mouse CD8[+] T cells. We find that low oxygen level diminishes CD8[+] T cell anti-viral response in vivo. We reveal that respiratory restriction inhibits ATP-dependent matrix processes that are critical for mitochondrial-biogenesis. This respiratory restriction-mediated effect could be rescued by TCA cycle re-stimulation, which yielded increased mitochondrial matrix-localized ATP via substrate-level phosphorylation. Finally, we demonstrate that the hypoxia-arrested CD8[+] T cell anti-viral response could be rescued in vivo through brief exposure to atmospheric oxygen pressure. Overall, these findings elucidate the detrimental effect of hypoxia on mitochondrial-biogenesis in activated CD8[+] T cells, and suggest a new approach for reducing viral infections in COPD.

*For correspondence:
michaelb@ekmd.huji.ac.il

**Competing interests:** The authors declare that no competing interests exist.

## Introduction

It is presently unclear exactly how CD8[+] T cell response is influenced by systemic oxygen restriction (SOR). This subject is difficult to investigate as it requires the identification of specific metabolic effects within the dynamic system of activated cells in a process of rapid transformation and rewiring (*MacIver et al., 2013*). This is important field of research, since hypoxemia, reduced blood oxygen saturation, and tissue hypoxia are associated with multiple respiratory and circulatory diseases, including chronic obstructive pulmonary disease (COPD) and congenital heart disease (*Kent et al., 2011*; *Kaskinen et al., 2016*; *O'Brien and Smith, 1994*). Reportedly, such patients also exhibit a higher prevalence of viral infections compared to healthy individuals (*Chaw et al., 2020*; *Kherad et al., 2010*).

Previous studies have examined how hypoxia affects cell fate determination in fully activated effector T cells (*Doedens et al., 2013*). Some have found that hypoxic conditions contribute to the formation of long-lasting effector cells (*Phan and Goldrath, 2015*; *Phan et al., 2016*). Other studies have demonstrated that respiratory restriction, mediated by inhibition of mitochondrial ATP synthase, arrests T cell activation (*Chang et al., 2013*). This is particularly interesting because activated T cells undergo an early shift in cell metabolism, in parallel to activation stimuli, switching to aerobic glycolysis to support their expansion and cytotoxic function (*Gubser et al., 2013*; *van der Windt*

*et al., 2013*). However, it remains unclear what mechanism underlies the inhibition of T cell activation under hypoxic conditions.

In the present study, we explored the effects of chronic systemic hypoxia on CD8$^+$ T cell response. To assess the possible effects of systemic hypoxia in vivo, we challenged mice with a lentivirus under conditions simulating COPD (*Yu et al., 1999*), and found that low oxygen availability diminished CD8$^+$ T cell response. Similarly, in vitro hypoxic conditions led to complete arrest of CD8$^+$ T cell response, but only marginally inhibited fully activated cells. To further characterize the metabolic mechanism underlying T cell arrest, we used the ATP synthase inhibitor oligomycin, which enables differentiation between indirect and direct effects of respiratory restriction. Incubation with oligomycin at different time-points post-stimuli revealed that after mitochondrial-biogenesis, at ~12 hr post-stimuli, activated CD8$^+$ T cells become independent of oxidative phosphorylation (OXPHOS). Next, to elucidate why CD8$^+$ T cells are sensitive to respiratory restriction prior to mitochondrial-biogenesis, we examined cytoplasmic response to respiratory restriction via metabolic profiling and p-AMPK analysis. This analysis revealed that respiratory restriction prior to mitochondrial-biogenesis had only a marginal effect on cytoplasmic function. Accordingly, the inhibition of mitochondrial ATP transport to the cytoplasm through genetic alteration or pharmacological treatment had little effect on CD8$^+$ T cell activation. In contrast, respiratory restriction prior to T cell mitochondrial-biogenesis, yielded an energetic crisis within the mitochondrial matrix, manifested by dysfunctional mitochondrial RNA processing and protein import. Moreover, oligomycin-treated CD8$^+$ T cells could be rescued using the proton ionophore FCCP, which uncouples the electron transport chain from ATP synthase. Finally, comparative metabolic profiling of oligomycin-treated activated T cells following uncoupler rescue, revealed significantly increased generation of mitochondrial matrix-localized ATP via mitochondria-localized substrate-level phosphorylation. Overall, these findings establish that during early activation, OXPHOS is required primarily to provide ATP for mitochondrial remodeling. By applying these insights to our in vivo model, we demonstrated that the detrimental effects of hypoxia may be alleviated by short oxygen resuscitation.

## Results

### Systemic chronic oxygen restriction inhibits CD8$^+$ T cell activation and response

Clinical chronic hypoxia is prevalent in multiple respiratory and circulatory diseases, and is associated with increased susceptibility to viral infections (*Chaw et al., 2020*; *Kherad et al., 2010*). Here we examined the effects of chronic hypoxia on CD8$^+$ T cell viral response by using a murine chronic hypoxia model (*Jain et al., 2016*; *Figure 1A*). Mice were intradermally primed in the ear pinna with an OVA-expressing lentivirus (Lv-OVA) (*Furmanov et al., 2013*; *Furmanov et al., 2010*). Twenty-four hours after the viral challenge, the mice were exposed to normal (atmospheric) or low (8%) oxygen levels for an additional 6 days. To evaluate the influence of oxygen levels on CD8$^+$ T cells' anti-viral response, we assessed the activation and proliferation status of OVA-associated CD8$^+$ T cells (TCR Vα2$^+$) from the deep cervical lymph nodes of the two experimental groups and from naïve mice. We quantified the relative abundance and total numbers of OVA-associated CD8$^+$ T cells presenting established in vivo activation markers, including elevation of hyaluronic acid receptor (CD44) and interleukin-2 receptor alpha chain (CD25), and reduction of L-selectin (CD62L). To evaluate the proliferation status of Ova-associated CD8$^+$ T cells, we also performed intracellular staining for Ki67.

Compared to the control group, the chronic hypoxia mice group showed a marked decrease in the ratios and the total numbers of CD62L$^-$ CD44$^+$ (*Figure 1B–C* and *Figure 1—figure supplement 1A–B*), CD62L$^-$ CD25$^+$ (*Figure 1D–E* and *Figure 1—figure supplement 1C*), and CD62L$^-$ CD44$^+$ CD25$^+$ (*Figure 1F*) Ova-associated CD8$^+$ T cells. Moreover, the phenotype of Ova-associated CD8$^+$ T cells from mice challenged under chronic hypoxia was similar to the untreated control. Mice challenged under chronic hypoxia exhibited a 10-fold decrease in Ki67-positive OVA-associated CD8$^+$ T cells compared to mice challenged under atmospheric oxygen levels (*Figure 1G–H*). These findings suggest that the induction of systemic chronic hypoxia in vivo disrupts the CD8$^+$ T cell response to viral infection.

To investigate the direct effect of hypoxia on T cell response , we activated spleen-derived lymphocytes in vitro for 72 hr, using a combination of the agonistic antibodies anti-CD3ε and anti-CD28,

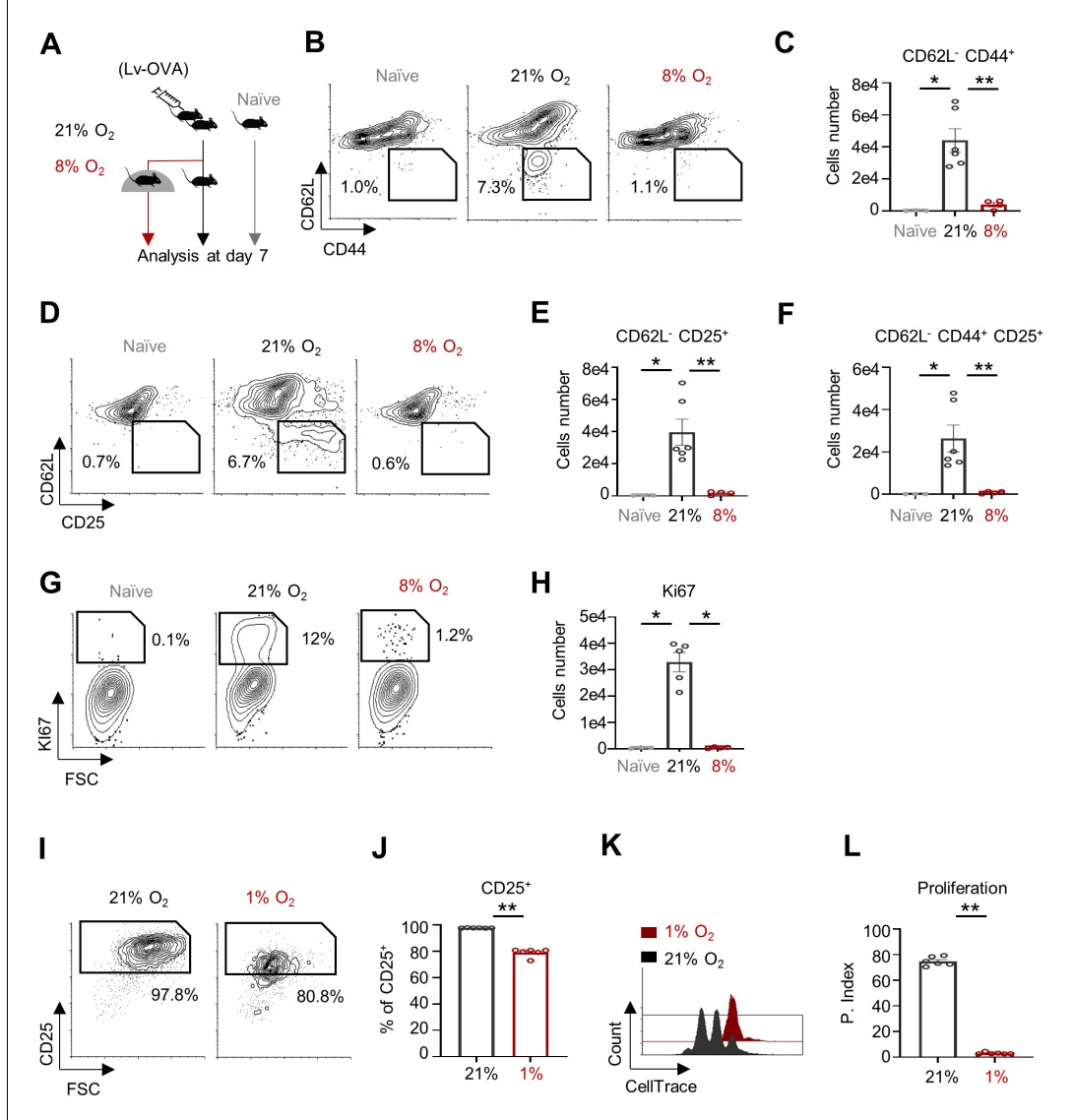

**Figure 1.** Systemic chronic oxygen restriction inhibits CD8$^+$ T cell activation and response. (**A**) Schematic of the experiment presented in panels **B-H**. C57BL6 mice were primed intradermally in the ear pinna with $5 \times 10^6$ transduction units (TU) of Lv-OVA or left untreated. Twenty-four hours following the viral challenge, mice were transferred to chambers for additional 6 days and kept under either 8% or 21% oxygen pressure. Extracted cells from the deep cervical lymph nodes were then analyzed by flow cytometry as follows: TCR Vα2$^+$, CD8$^+$ T cells from naïve mice (left/gray) or Lv-OVA challenged mice that were kept under either atmospheric oxygen pressure (21% O$_2$) (middle/black) or 8% oxygen pressure (right/red). (naïve n = 3 biological replicates, activated n = 6 in each group). (**B**) Representative flow cytometry plots of CD44 vs. CD62L, numbers indicate the frequencies of CD44$^+$ CD62L$^-$ cells. (**C**) Bar graph quantification of TCR Vα2$^+$, CD8$^+$, CD44$^+$, CD62L$^-$ cells (P value * 0.0238, ** 0.0095). (**D**) Same as in **B**, focusing on CD25 vs. CD62L. (**E**) Same as in **C** focusing on TCR Vα2$^+$, CD8$^+$, CD62L$^-$ CD25$^+$ cells. (P value * 0.0238, ** 0.0095). (**F**) Same as in **C**, focusing on TCR Vα2$^+$, CD8$^+$, CD62L$^-$ CD25$^+$, CD44$^+$ cells. (P value * 0.0238, ** 0.0095). (**G**) Representative flow cytometry plots of Ki67 vs. FSC gated on TCR Vα2$^+$, CD8$^+$, CD25$^+$ T cells. (**H**) Bar graph quantification of TCR Vα2$^+$, CD8$^+$, CD25$^+$, Ki67$^+$ cells. (P value * 0.0357, * 0.0159). (**I-L**) CellTrace-labeled splenocytes were activated with anti-CD3/28 for 72 hr under either 1% or 21% of oxygen. (n = 5 biological replicates). (**I**) Representative flow cytometry plots of FSC vs. CD25 gated on CD8$^+$ T cells. Numbers indicate the frequencies of CD25$^+$ cells. (**J**) Bar graph summarizes results in **I**. (P value * 0.0022). (**K**) Representative Flow cytometry overlay histogram of CellTrace intensity gated on CD8$^+$ T cells. (**L**) Bar graph summarizes results in **K** as Proliferation Index (P. Index). (P value * 0.0022). Statistical method, non-parametric Mann–Whitney test, mean ± s.e.m.

The online version of this article includes the following figure supplement(s) for figure 1:

**Figure supplement 1.** Systemic chronic oxygen restriction inhibits CD8$^+$ T cell activation and response.

in an oxygen-deficient (1% $O_2$) environment. In accordance with our in vivo findings and previous reports (*Chang et al., 2013*), naïve CD8+ T cells ($T_n$) activated under hypoxic conditions exhibited reduced levels of the in vitro activation marker CD25 (*Figure 1I–J*) and diminished proliferative capacity (*Figure 1K–L*) compared to cells activated under atmospheric oxygen levels. Finally, to examine the relevance of our model system to human immunity, we activated human CD8+ T cells under either atmospheric oxygen levels or hypoxic conditions. As expected, and similar to the findings in mouse cells, human CD8+ T cells activated under hypoxic conditions exhibited a marked decrease in surface expression of CD25, and proliferative capacity (*Figure 1—figure supplement 1D–G*). Together, our findings demonstrate that CD8+ T cell activation is compromised during systemic chronic hypoxia.

## Following mitochondrial remodeling, activated CD8+ T cells become tolerant to inhibition of OXPHOS

Next, we aimed to investigate the oxygen-dependent mechanisms governing T cell transition from the naïve to the activated state. It was previously demonstrated that although hypoxia negatively impacts naïve T cell proliferation after activation, it does not significantly impact the proliferation or function of effector T cells. Moreover, some reports show that hypoxia can actually improve effector T cell functions in vivo (*Doedens et al., 2013*; *Gropper et al., 2017*; *Makino et al., 2003*; *Vuillefroy de Silly et al., 2016*; *Xu et al., 2016*). These findings suggest that T cells acquire hypoxia tolerance during their activation. Thus, defining the metabolic alterations between hypoxia-sensitive and hypoxia-resistance T cells could elucidate the inhibitory effects of systemic hypoxia on CD8+ T cell activation. To characterize this metabolic transition, CD8+ T cells were activated in vitro and subjected to hypoxia at early (5 hr) and late (18 hr) time-points post-stimuli (*Figure 2A*). Notably, CD8+ T cells exposed to hypoxia at the early time-point following activation exhibited impaired elevation of CD25 expression and decreased proliferative capacity. These impairments were partially prevented in cells transferred to hypoxic conditions at the late time-point following activation (*Figure 2B–E*).

To investigate the hypoxia-mediated inhibitory effect, we used the ATP synthase-specific inhibitor oligomycin, which partially mimics the effect induced by hypoxia , imposing cellular respiratory restriction (*Chang et al., 2013*; *Solaini et al., 2010*; *Sgarbi et al., 2018*). We utilized oligomycin because it provides a simple experimental system to test the immediate effect of respiratory restriction under multiple conditions. Importantly, it enables differentiation between indirect effects mediated by inhibition of the electron transport chain and the TCA cycle (*Martínez-Reyes et al., 2016*) versus the direct effects caused by reduced mitochondrial ATP (*Lee and O'Brien, 2010*). Oligomycin titration assays confirmed that 60 nM oligomycin had stable and significant effect on CD8+ T cell respiration, activation, and proliferation (*Figure 2—figure supplement 1A–G*).

To pin-point the development of tolerance to respiratory restriction, we examined CD8+ T cell response following oligomycin treatment at multiple time-points post-stimuli (*Figure 2F*). Activated CD8+ T cells treated with oligomycin at time-points earlier than 9 hr post-stimuli (T-Early) showed decreased CD25 expression and proliferation. However, when oligomycin was added at time-points later than 12 hr post-stimuli (T-Late), we observed a significant increase of CD25 expression and proliferation (*Figure 2G–J*). Taken together, these observations suggest a gradual metabolic rewiring process that promotes the development of a metabolic bypass of respiratory restriction in T-Late.

Glycolysis, the degradation of glucose to pyruvate/lactate, allows cellular ATP generation independent of oxygen concentrations (*Lunt and Vander Heiden, 2011*). Here we tested whether the cellular capacity to perform glycolysis is correlated with the acquisition of tolerance to respiratory restriction during CD8+ T cell activation. To account for variances in glycolysis (*Gubser et al., 2013*; *van der Windt et al., 2013*), we assessed the extracellular acidification rate (ECAR) of $T_n$, T-Early, and T-Late using the seahorse methodology. Interestingly, both basal ECAR (without oligomycin treatment) and maximal ECAR (with oligomycin treatment) were comparable between T-Early and T-Late, and these rates were elevated with respect to $T_n$ (*Figure 2—figure supplement 2A–C*). To test whether the glucose-uptake rates differed between early and late activation, we incubated CD8+ T cells with the fluorescent glucose-uptake probe 2-deoxy-2-D-glucose (2-NBDG) at different time-points post-stimuli, and then used flow cytometry to analyze their glucose uptake. The glucose-uptake rate did not substantially differ between $T_n$ and the CD8+ T cells activated for 6, 9, or 12 hr

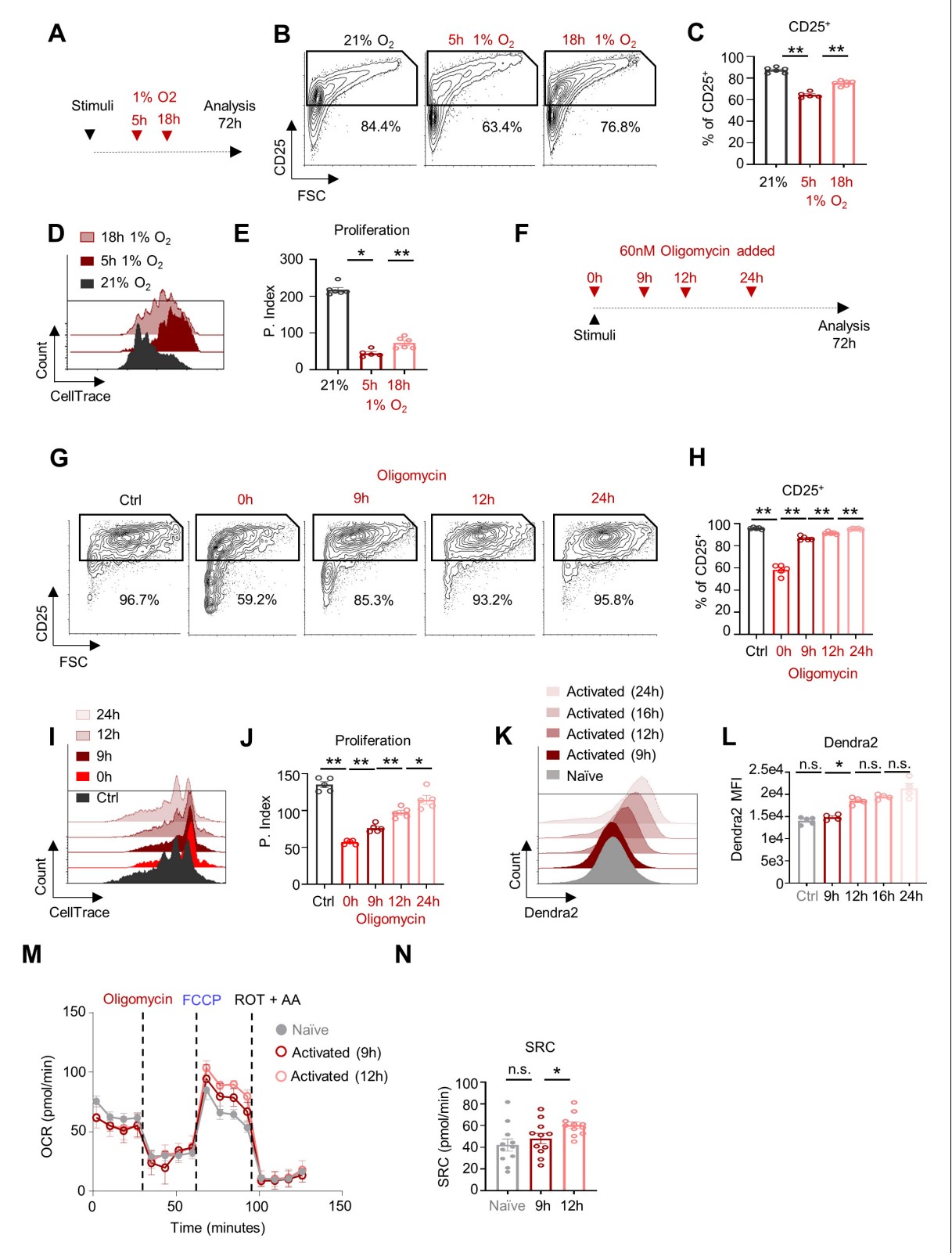

**Figure 2.** Following mitochondrial remodeling, activated CD8+ T cell become tolerant to inhibition of OXPHOS. (**A**) Schematic of experiment presented in panels **B-E**. CellTrace-labeled splenocytes were activated with anti-CD3/28 and transferred to chambers containing 1% $O_2$ for 5 (middle/red), or 18 hr (right/pink). Cells were then transferred to atmospheric oxygen pressure. Control cells were left in atmospheric oxygen pressure (21% $O_2$), (left/gray). Seventy-two hours post activation, cells were analyzed by flow cytometry. (n = 5 biological replicates). (**B**) Representative flow cytometry plots of FSC vs.

*Figure 2 continued on next page*

*Figure 2 continued*

CD25 gated on CD8$^+$ T cells. Numbers indicate the frequencies of CD25$^+$ cells. (C) Bar graph summarizes results in B. (P value ** 0.0043, ** 0.0043). (D) Representative flow cytometry overlay histogram of CellTrace intensity gated on CD8$^+$ T cells. (E) Bar graph summarizes results in D as Proliferation Index. (P value ** 0.0043, * 0.0173). (F) Schematic of experiment presented in panels G-J. CellTrace-labeled splenocytes were stimulated using anti-CD3/28 and treated with 60 nM oligomycin at the indicated time points post activation. Seventy-two hours post activation cells were analyzed by flow cytometry. (n = 5 biological replicates). (G) Representative flow cytometry plots of FSC vs. CD25 gated on CD8$^+$ T cells of control cells (untreated with oligomycin) or cells that were treated with oligomycin at the indicated time points after activation. Numbers indicate the frequencies of CD25$^+$ cells. (H) Bar graph summarizes results in G. (P value ** 0.0079 ** 0.0079 ** 0.0079 ** 0.0079). (I) Representative flow cytometry overlay histogram of CellTrace intensity gated on CD8$^+$ T cells from either control cells (front) or cells that were treated with oligomycin at the indicated time after activation. (J) Bar graph summarizes results in I as Proliferation Index. (P value ** 0.0079 ** 0.0079 ** 0.0079 * 0.0317). (K) Flow cytometry histogram overlay plot of Dendra2 fluorescence intensity gated on the CD8$^+$ T cell population of naïve cells or cells that were stimulated using anti-CD3/CD28, for 9, 12, 16 or 24 hr from spleens of mito-Dendra2 mice. (L) Bar graph summarizes results in K. (n = 5 biological replicates). (P value * 0.0286). (M) Mouse splenocytes were stimulated using anti-CD3/CD28 for 9 (pink), 12 (red) hours or left untreated (Naïve- gray). CD8$^+$ T cells were then isolated and assayed, by seahorse XF24, for Oxygen Consumption Rate (OCR) following consecutive injections of oligomycin, FCCP, and rotenone plus antimycin (R+A). (n = 5 biological replicates in each group). (N) Bar graph summarizes the spare respiratory capacity (SRC- maximal OCR after FCCP treatment) of the experiment present in M. (P value * 0.0197) Statistical method, non-parametric Mann–Whitney test, mean ± s.e.m.

The online version of this article includes the following figure supplement(s) for figure 2:

**Figure supplement 1.** Oligomycin inhibits CD8$^+$ T cell respiration, activation, and proliferation.
**Figure supplement 2.** Acquisition of respiratory restriction tolerance is not correlated with increased glycolytic activity.

(*Figure 2—figure supplement 2D*). At a later stage of the activation, 24 hr post-stimuli, we observed a considerable increase in the amount of glucose uptake.

Metabolic analysis of naïve, T-Early, and T-Late cells revealed significantly altered concentrations of key glycolysis-related metabolites in both T-Early and T-Late cells with respect to T$_n$ (*Figure 2—figure supplement 2E–F*). Specifically, T-Early exhibited higher levels of intracellular glucose 6-phosphate (G6P), and reduced glucose levels (*Figure 2—figure supplement 2E*). Similarly, the secreted lactate concentration was substantially elevated in T-Early compared to T$_n$ (*Figure 2—figure supplement 2F*). In contrast, glycolysis-related metabolites did not significantly differ between T-Late and T-Early cells (*Figure 2—figure supplement 2E–F*). Collectively, our observations demonstrate that stimulated CD8$^+$ T cells exhibited a marked increase of glycolytic metabolism at least 9 hr before they acquired tolerance to respiratory restriction. These findings indicate that the development of tolerance to respiratory restriction is not correlated with increased glycolytic activity.

Previous reports show that mitochondria undergo robust biogenesis and extensive metabolic rewiring at ~12 hr after T cell activation (*Rambold and Pearce, 2018*; *Ron-Harel et al., 2016*). To determine whether mitochondrial-biogenesis correlates with the acquisition of tolerance to respiratory restriction, we measured the kinetics of mitochondrial-biogenesis during CD8$^+$ T cell activation in our model system. Stimulated CD8$^+$ T cells derived from mitochondria-labeled mtDendra2 mice (*Pham et al., 2012*) showed a substantial increase of mitochondrial mass at 12 hr post-stimuli, correlating with CD8$^+$ T cell acquisition of tolerance to respiratory restriction (*Figure 2K–L*). Likewise T-Late cells exhibited a significant increase in spare respiratory capacity in comparison to T-Early (*Figure 2M–N*). Thus, upon CD8$^+$ T cell activation, the development of tolerance to respiratory restriction is correlated with a gain of mitochondrial biomass linked to mitochondrial rewiring (*Ron-Harel et al., 2016*).

## Respiratory restriction has only a marginal effect on cytoplasmic function during early activation

The inhibitory effect mediated by respiratory restriction during CD8$^+$ T cell activation could be caused by reduced mitochondrial ATP (*Lee and O'Brien, 2010*) and increased AMP-related signaling (*Araki et al., 2009*; *Pearce et al., 2009*; *Araki et al., 2009*; *Pearce et al., 2009*). Therefore, we next investigated whether respiratory-restriction results in increased AMP-related signaling. To assess how respiratory restriction influences the levels of different phospho-nucleotides, we examined the metabolic profiles of oligomycin-treated T-Early and T-Late cells compared to untreated controls. As expected, oligomycin treatment yielded a marked increase of mono/di-phospho-nucleotides at the expense of tri-phospho-nucleotides in T-Early cells (*Figure 3A*). A similar effect was

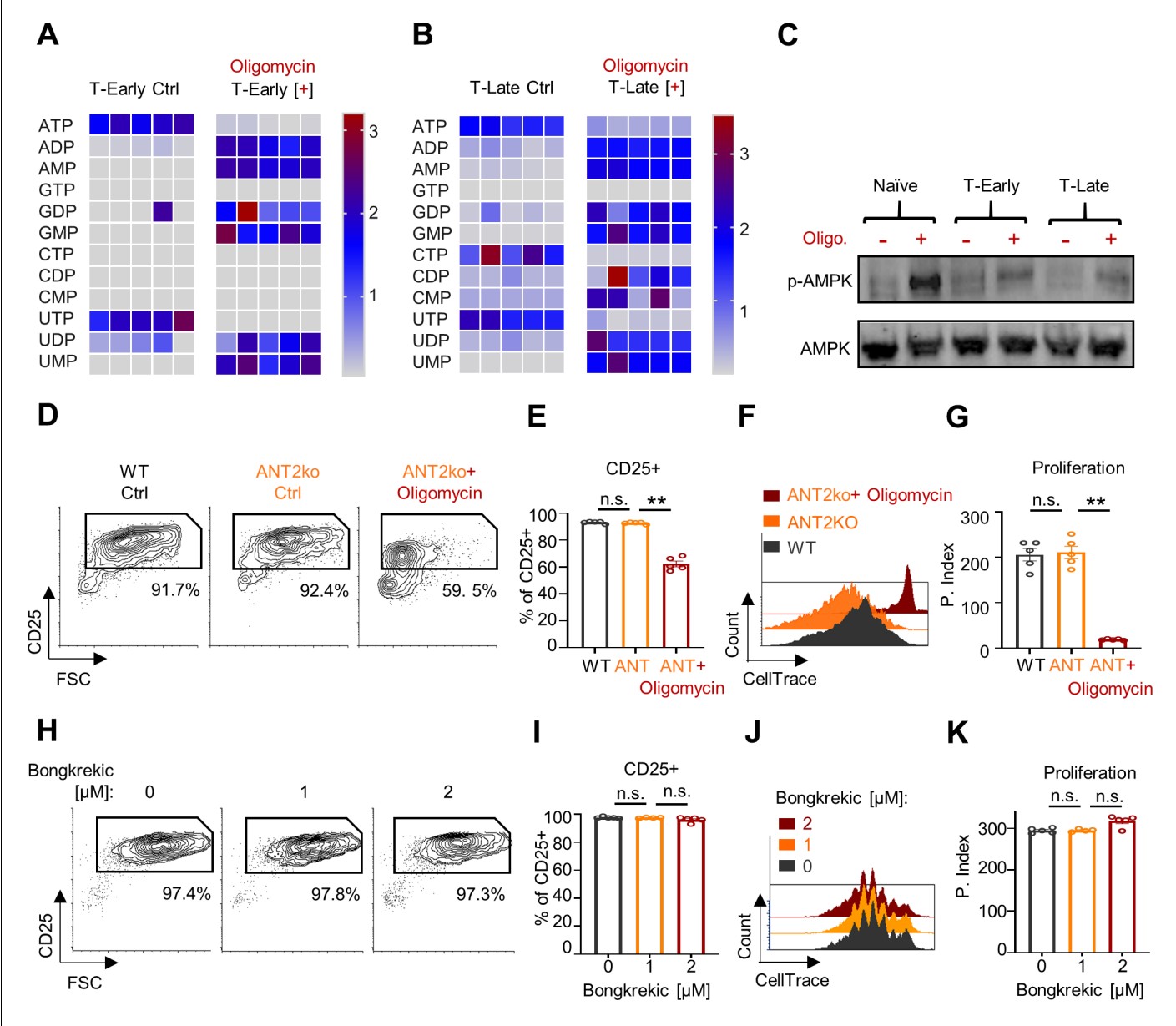

**Figure 3.** Respiratory restriction has only marginal effect on cytoplasmic function during early activation. (**A**) Heatmap showing relative amounts of key energy-related metabolites (as indicated in the figure) extracted from CD8+ T cells activated for 5 hr using anti-CD3/28 (T-Early) oligomycin treated or untreated (Ctrl). (**B**) Same as in **A**, extracted from CD8+ T cells activated for 24 hr using anti-CD3/28 (T-Late). (**C**) Splenocytes were stimulated using anti-CD3/28 for 9 hr (T-Early), 12 hr (T-Late) or left untreated (Naïve- gray). Cells were then treated with 300 nM oligomycin or left untreated for 1 hr. Protein extract from isolated CD8+ T cells from all samples were then subjected for immunoblot analysis using anti p-AMPKα or anti AMPKα. (n = 3 experiments). (**D-G**) CellTrace-labeled splenocytes from WT or LCK-cre/*Slc25a5*floxp (ANT2ko) mice were stimulated using anti-CD3/CD28, with or without 60 nM oligomycin. Seventy-two hours post activation cells were analyzed by flow cytometry (n = 5 biological replicates). (**D**) Representative Flow cytometry plots of FSC vs. CD25 gated on CD8+ T cells from wild-type mice (WT- left) or ANT2ko mice untreated or treated with 60 nM oligomycin (middle and right panels respectively). Numbers indicate the frequencies of CD25+ cells. (**E**) Bar graph summarizes results in D. (P value ** 0.0079). (**F**) Representative Flow cytometry overlay histogram of CellTrace intensity gated on CD8+ T cells from either WT cells (front) or ANTko cells that were untreated (middle) or treated with oligomycin (back). (**G**) Bar graph summarizing the results in F as proliferation index. (P value ** 0.0079). (**H-K**) CellTrace-labeled splenocytes from WT mice were stimulated using anti-CD3/CD28, in the presence of the indicated concentrations of the pan-ANT inhibitor, bongkrekic acid. Seventy-two hours post activation cells were analyzed by flow cytometry analysis. (n = 5 biological replicates). (**H**) Representative Flow cytometry plots of FSC vs. CD25 gated on CD8+ T cells from WT splenocytes untreated or treated with the indicated concentrations of bongkrekic acid. Numbers indicate the frequencies of CD25+ cells. (**I**) Bar graph summarizing the results in H. (**J**) Representative Flow cytometry overlay histogram of CellTrace intensity gated on CD8+ T cells that were either untreated or treated with the indicated concentrations of bongkrekic acid. (**K**) Bar graph summarizing the results in J as proliferation index. Statistical method, non-parametric Mann–Whitney test, mean ± s.e.m.
*Figure 3 continued on next page*

Figure 3 continued

The online version of this article includes the following figure supplement(s) for figure 3:

**Figure supplement 1.** Inhibition of ANT2 leads to increased mitochondrial membrane polarization in CD8[+] T cells.

observed in hypoxia-resilient T-Late cells exposed to oligomycin (*Figure 3B*), suggesting that activated CD8[+] T cells may function under increased AMP levels.

We further evaluated the levels at which oligomycin-induced changes in the phospho-nucleotide profile affects cellular energy sensing, by examining the activation levels of AMP-related signaling. To this end, we measured the level of phosphorylated AMP-activated protein kinase (p-AMPK) as a marker for AMPK activation, which is a cytoplasmic sensor for energy homeostasis, in both T-Early and T-Late cells. Treatment with oligomycin, which is the typical positive control for AMPK activation, markedly increased the p-AMPK levels in treated naïve T cells. Interestingly following activation oligomycin only marginally increased the level of p-AMPK in respect to untreated activated control. Importantly, the response to oligomycin was comparable between T-Early and T-Late cells (*Figure 3C*). Together, these results suggest that despite the important role of AMPK signaling in T cell metabolic adaptation (*Blagih et al., 2015*), it is not correlated with the inhibitory effects mediated by respiratory restriction in early activation. Further, the respiratory dependence during early T cell activation is not caused by an altered cellular response to reduced AMP levels in the cytoplasm.

Our findings to this point imply that activated CD8[+] T cells have a unique capacity to avoid p-AMPK signaling in the presence of elevated AMP levels. To support these findings, we tested how depleting mitochondrial ATP from CD8[+] T cells' cytoplasmic compartment affects their activation. To investigate this possibility, we generated T cell-specific adenine nucleotide translocator 2 (ANT2 is encoded by the *Slc25a5* gene) knockout mice (refer to as ANT2ko) (*Cho et al., 2017*; *Cho et al., 2015*). ANT2 is the dominant ADP/ATP translocator in murine CD8[+] T cells, constituting approximately 90% of the total ANT protein (*Figure 3—figure supplement 1A*). Importantly, ANT2ko CD8[+] T cells displayed substantially increased mitochondrial membrane polarization (*Figure 3—figure supplement 1B*), indicating a decreased matrix ADP concentration.

To determine whether T cell-specific ANT2 deletion affected T cell activation, we examined the ANT2ko-derived CD8[+] T cells' response to stimuli. Surprisingly, ANT2-deficient T cells exhibited intact activation-induced CD25 expression (*Figure 3D–E*) and robust proliferative capacity (*Figure 3F–G*). Notably, the ANT2ko-derived CD8[+] T cells were still sensitive to respiratory restriction during early activation (*Figure 3D–G*).

T cell-specific ANT2 deletion provides a model of chronic restriction of mitochondrial ATP in the cytoplasm. To account for any compensatory effects that may have developed in these mice over time, and to observe the influence of acute mitochondrial ATP restriction to the cytoplasm, we treated activated CD8[+] T cells with increasing doses of the pan-ANT inhibitor bongkrekic acid (*Anwar et al., 2017*). CD8[+] T cells stimulated in the presence of bongkrekic acid, at concentrations that increase mitochondrial membrane polarization (*Figure 3—figure supplement 1C*), exhibited an increase of CD25 surface expression (*Figure 3H–I*) and proliferation patterns (*Figure 3J–K*) that were similar to the untreated control group. These key observations illustrate that ATP generated by mitochondrial respiration is not required for cytoplasmic function of activated CD8[+] T cell. Furthermore, our results suggest that an upstream respiratory-restriction-coupled effect is a limiting factor underlying CD8[+] T cells' sensitivity to respiratory restriction during early activation.

## Respiratory restriction leads to energetic crisis within the matrix compartment in early activated CD8[+] T cells

Mitochondrial-biogenesis and rewiring are critical checkpoints in T cell activation (*Ron-Harel et al., 2016*; *Rambold and Pearce, 2018*). These cellular processes rely on the availability of matrix-bond ATP, which is generated by substrate-level phosphorylation, the metabolism of succinyl-CoA to succinate in the TCA cycle (*Schwimmer et al., 2005*; *Chinopoulos et al., 2010*; *Bochud-Allemann and Schneider, 2002*). Therefore, we next examined whether respiratory restriction affects mitochondrial-biogenesis via an upstream effect. As expected, mtDendra2-derived T-Early cells transferred to a hypoxic chamber exhibited reduced CD25 expression (*Figure 4A–B*). Importantly, T-Early cells from the hypoxia group showed significantly reduced mtDendra2 expression (*Figure 4C–D*).

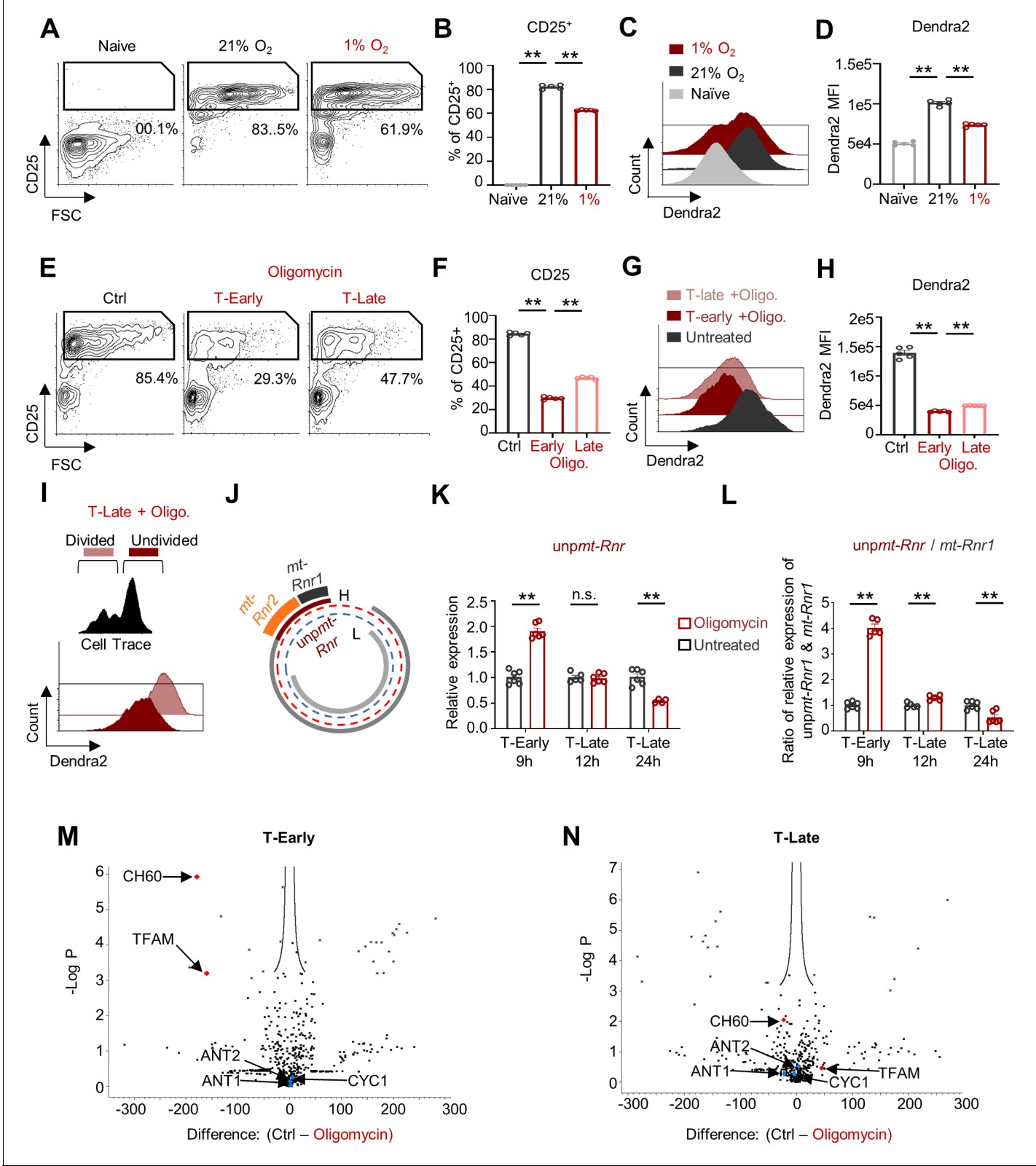

**Figure 4.** Respiratory restriction leads to energetic crisis within the matrix compartment in early activated CD8[+] T cell. (A-D) Splenocytes from mito-Dendra2 mice were activated with anti-CD3/28 for 5 hr and then transferred to a chamber containing 1% O$_2$ or left in atmospheric oxygen pressure (21% O$_2$). Twenty-four hours post activation cells were analyzed by flow cytometry. (A) Representative flow cytometry plots of FSC vs. CD25 gated on CD8[+] T cells. Numbers indicate the frequencies of CD25[+] cells. (B) Bar graph summarizing the results in A. (P value ** 0.0079 ** 0.0079). (C)

*Figure 4 continued on next page*

*Figure 4 continued*

Representative Flow cytometry histogram overlay plot of Dendra2 fluorescence intensity gated on the CD8$^+$ T cells from Naïve (front), activated in 21% O$_2$ (middle), or activated in 1% O$_2$ (back) cells. (D) Bar graph summarizing the results in C, Dendra2 mean fluorescence intensity (MFI). (P value ** 0.0079 ** 0.0079). (E-H) Splenocytes from mito-Dendra2 mice were activated with anti-CD3/28 and treated with 60 nM oligomycin at 9 hr (T-Early) or 12 hr (T-Late) following activation. Twenty-four hours post activation cells were analyzed by flow cytometry. (E) Representative flow cytometry plots of FSC vs. CD25 gated on CD8$^+$ T cells. Numbers indicate the frequencies of CD25$^+$ cells. (F) Bar graph summarizing the results in E. (P value ** 0.0079 ** 0.0079). (G) Representative Flow cytometry histogram overlay plot of Dendra2 fluorescence intensity gated on the CD8$^+$ T cells from either oligomycin untreated (front), oligomycin treated T-Early (middle) or oligomycin treated T-Late (back). (P value ** 0.0079 ** 0.0079). (H) Bar graph summarizing the results in G, Dendra2 mean fluorescence intensity (MFI). (P value ** 0.0079 ** 0.0079). (I) CellTrace-labeled splenocytes from mito-Dendra2 mice were activated with anti-CD3/28 and treated with 60 nM oligomycin 12 hr (T-Late) after activation. Seventy-four hours post activation cells were analyzed by flow cytometry. Figure shows Representative Flow cytometry histogram overlay plot of Dendra2 fluorescence intensity gated on the undivided (highest CellTrace intensity- front) and divided (low CellTrace intensity) CD8$^+$ T cell populations. (J) Schematic of typical mitochondrial transcription, describing the transcription of ribosomal polycistronic mitochondrial RNA (unp*mt-Rnr*, dark red line) from the mitochondrial DNA heavy strand (H, red dotted line), and its processing to *mt-Rnr1* (bold black line) and *mt-Rnr2* (bold orange line). (K-L) Splenocytes were stimulated using anti CD3/CD28 for 9, 12 and 24 hr. Cells were then treated with oligomycin for one hour or left untreated. Total RNA was extracted from isolated CD8$^+$ T cells and assayed using qRT-PCR. Primers were designed to amplify either the unprocessed *mt-Rnr* transcript (unp*mt-Rnr*) or its two processed products, *mt-Rnr1* and *mt-Rnr2*. (K) Relative expression of the unp*mt-Rnr*. (P value ** 0.0022 ** 0.0095). (L) Ratio between the relative expression of unp*mt-Rnr* and *mt-Rnr1*. (n = 6 biological replicates). (P value ** 0.0043 ** 0.0079 ** 0.0087). (M-N) Splenocytes were stimulated using anti-CD3/CD28 for 9 or 12 hr. One hour prior to CD8$^+$ T cells isolation, cells were treated with oligomycin or left untreated. Protein extracts from isolated CD8$^+$ T cells were then subjected to immunoprecipitation (IP) using anti-ubiquitin antibody. IP extracts were analyzed by MS focusing on mitochondrial proteins and mitochondrial leader peptides. (n = 3 biological replicates). (M) Volcano plot of precipitated proteins detected by the MS analysis in oligomycin treated and untreated samples (9 hr, T-Early). (N) Volcano plot of precipitated proteins detected by the MS analysis in oligomycin treated and untreated samples (12 hr, T-Late). Statistical method, (A-L) non-parametric Mann–Whitney test, mean ± s.e.m, (M-N) False Discovery Rate (FDR) P value < 0.05.

The online version of this article includes the following figure supplement(s) for figure 4:

**Figure supplement 1.** Respiratory restriction induces an energetic crisis within the matrix compartment in early activated CD8$^+$ T Cell.

---

Similarly, oligomycin treatment during early activation of CD8$^+$ T cells abrogated activation (*Figure 4E–F*) and inhibited the increase of mitochondrial mass that was observed in control mtDendra2-derived CD8$^+$ T cells (*Figure 4G–H*). Interestingly, we observed substantially higher mtDendra2 expression in proliferating T-Late cells compared to undivided T-Late cells (*Figure 4I*), suggesting that respiratory restriction inhibits activation by disrupting mitochondrial-biogenesis.

Next, we investigated whether respiratory restriction leads to an energetic crisis within the mitochondrial matrix. We applied a functional approach to determine whether acute respiratory restriction disturbs ATP-dependent processes within the matrix (*Supplementary file 1*). We focused on two processes; (1) protein import, in which mitochondria-localized pre-proteins contain a leader sequence that is cleaved and removed upon matrix entry (*Pfanner et al., 2019*; *Wiedemann and Pfanner, 2017*; *Chacinska et al., 2009*-), and (2) processing of the polycistronic mitochondrial RNA that encodes the 12S and 16S ribosomal RNAs (Rnr1 and Rnr2) (*Tu and Barrientos, 2015*; *Wang et al., 2010*; *Buck et al., 2016*). To evaluate whether respiratory restriction promotes disturbance in matrix-localized RNA processing, we quantified the relative expression levels of unprocessed Rnr polycistronic mitochondrial RNA (unpmt-Rnr), as well as the ratio between unpmt-Rnr and its processed products, Rnr1 and Rnr2 (*Figure 4J*; *Rackham et al., 2016*). Oligomycin treatment yielded markedly increased unpmt-Rnr levels in T-Early cells but not in T-Late cells (*Figure 4K*). Accordingly, we found that the ratios between unpmt-Rnr and its cleaved products, Rnr1 or Rnr2, were further increased in oligomycin-treated T-Early cells compared to T-Late cells (*Figure 4L* and *Figure 4—figure supplement 1A*).

During matrix ATP deficiency, the protein import machinery cannot pull nuclear-encoded matrix proteins. This protein import disturbance causes matrix proteins to misfold in the cytoplasm, leading to their degradation via the ubiquitin-proteasome pathway (*Chacinska et al., 2009*; *Figure 4—figure supplement 1B*). Therefore, we next assessed whether respiratory restriction also resulted in the accumulation of ubiquitinated mitochondrial matrix proteins and, specifically, ubiquitinated matrix pre-proteins. T-Late and T-Early cells were treated with oligomycin or left untreated for 1 hr. Then protein extracts from all samples were subjected to immunoprecipitation using anti-ubiquitin antibody, and assayed using mass spectrometry. As expected, in T-Early cells, acute oligomycin treatment significantly increased the amounts of at least two central matrix proteins when compared to controls without oligomycin treatment. Specifically, T-Early cells exhibited increased abundances of

ubiquitinated mitochondrial transcription factor A (TFAM) and the CH60 chaperone, which plays a role in the folding and assembly of newly imported proteins in the mitochondria. In line with the partial tolerance to oligomycin observed during late activation, oligomycin-treated T-Late cells showed no substantial increase in ubiquitinated matrix proteins compared to untreated controls (*Figure 4M-N* and its source data). Importantly, in all groups, the oligomycin-treated samples and controls did not significantly differ in the amounts of the mitochondrial inner-membrane proteins ANT2, ANT1, and CYC1, which do not require matrix ATP for mitochondrial localization. Furthermore, in samples from oligomycin-treated T-Early cells, we detected leader peptides of several mitochondrial proteins whose mitochondrial import depends on matrix ATP (*Figure 4—figure supplement 1C-D*). In contrast, no relevant leader peptides were detected in any of the untreated samples or in the oligomycin-treated T-Late samples (*Figure 4—figure supplement 1C-D*). Taken together, these results reveal a matrix-specific energetic crisis following oligomycin mediated respiratory restriction during early CD8$^+$ T cell activation, and suggest that inhibition of TCA cycle and substrate-level phosphorylation may be the central inhibitory mechanism of respiratory restriction.

## FCCP treatment rescues respiratory-restricted CD8$^+$ T cells by stimulating matrix-localized substrate-level phosphorylation, elevating ATP, and reducing AMP/GMP concentrations

TCA-linked substrate-level phosphorylation is thought to fuel mitochondrial matrix activity, while ATP synthase-derived ATP is exported to the cytoplasm (*Schwimmer et al., 2005*; *Bochud-Allemann and Schneider, 2002*). Oligomycin may indirectly lead to TCA cycle congestion, accumulation of intermediate metabolites, and blockade of matrix-based substrate-level phosphorylation. In this case, the addition of uncouplers to respiratory ATP-deprived T-Early cells could rescue the activation phenotype via TCA cycle stimulation (*Figure 5—figure supplement 1*). Therefore, we first attempted to rescue oligomycin-treated CD8$^+$ T cells by uncoupling the respiratory chain using an effective concentration of trifluoromethoxy carbonylcyanide phenylhydrazone (FCCP), which is a potent uncoupler of oxidative phosphorylation in mitochondria. T-Early cells were treated with oligomycin, FCCP, both oligomycin and FCCP, or were left untreated. As expected, oligomycin treatment during early activation arrested CD8$^+$ T cell proliferation. FCCP treatment alone slightly inhibited CD8$^+$ T cell proliferation compared to control. Strikingly, treatment of stimulated CD8$^+$ T cells with both oligomycin and FCCP led to an almost complete rescue of CD25 expression (*Figure 5A-B*) and proliferation (*Figure 5C-D*) compared to the cells treated with only oligomycin. These key observations demonstrate that uncoupling the respiratory chain from ATP synthase rescues the respiratory-restricted T-Early cells, suggesting that the inhibitory mechanism that follows respiratory restriction is linked to a decrease in ATP concentration in the matrix compartment.

The release of TCA cycle inhibition may allow respiratory-restricted cells to recover their matrix ATP via substrate-level phosphorylation, specifically through the conversion of succinyl-CoA to succinate. Accordingly, it would be expected that FCCP treatment would allow respiratory-restricted T-Early cells to replenish their mitochondria with ATP, thus rescuing their matrix energy levels despite the inhibition of ATP synthase. We investigated this possibility by examining the levels of mono/di/tri-phosphonucleotides using metabolomics analysis. As expected, T-Early cells treated with oligomycin and FCCP exhibited significantly higher ATP and lower AMP/GMP concentrations compared to controls treated with only oligomycin (*Figure 5E-F*). Given the ATP synthase blockade, the increase of cellular ATP and reductions of AMP and GMP may be primarily attributed to matrix-bound substrate-level phosphorylation.

Finally, to confirm that oligomycin treatment caused a surge in TCA cycle intermediates, we analyzed the metabolic profiles of T-Early cells that were incubated in media containing labeled $^{13}$C-glutamine following treatment with oligomycin or with oligomycin plus FCCP, compared to without treatment. In line with our hypothesis, respiratory restriction yielded marked increases of several key TCA cycle metabolites—including succinate, malate, and citrate—compared to control (*Figure 5G* and its source data). Additionally, relative to controls, oligomycin treatment led to increased levels of glutamate and aspartate, which are linked to the TCA cycle via the Gaba Shunt and the malate-aspartate shuttle, respectively (*Figure 5G*). Importantly, the addition of FCCP to oligomycin-treated T-Early cells reduced the signal levels of all TCA-linked intermediates, indicating stimulation of the TCA cycle (*Figure 5G*).

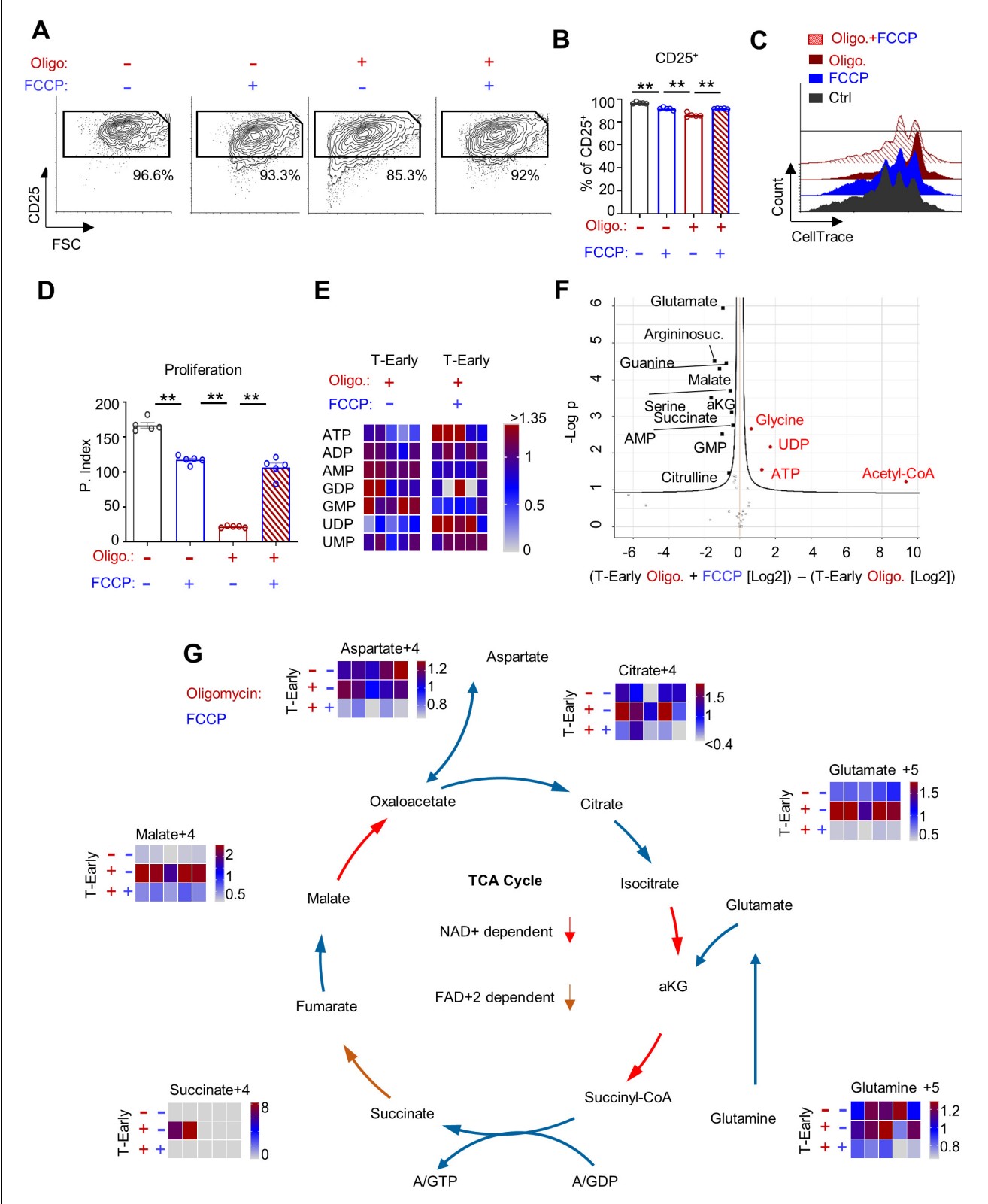

**Figure 5.** FCCP treatment rescues respiratory-restricted CD8[+] T cells by stimulating matrix-localized substrate-level phosphorylation, elevating ATP, and reducing AMP/GMP concentrations. (A-D) CellTrace-labeled splenocytes were stimulated using anti-CD3/28. Nine hours post activation cells were left untreated or treated with 1 μM FCCP, 60 nM oligomycin or with a combination of FCCP and oligomycin. Seventy-two hours post activation cells were analyzed by flow cytometry. (n = 5 biological replicates). (**A**) Representative flow cytometry plots of FSC vs. CD25 gated on CD8[+] T cells from

*Figure 5 continued on next page*

Figure 5 continued

untreated cells (left), FCCP treated (second from left), Oligomycin treated (third from left), or Oligomycin and FCCP treated (left). Numbers indicate the frequencies of CD25$^+$ cells. (B) Bar graph summarizing the results in B. (P value ** 0.0079 ** 0.0079 ** 0.0079). (C) Representative flow cytometry overlay histogram of CellTrace intensity gated on CD8$^+$ T cells from untreated cells (black), FCCP treated (blue), oligomycin treated (dark red-brown), or oligomycin and FCCP treated (red stripes). (D) Bar graph summarizing the results in DC as proliferation index. (P value ** 0.0079 ** 0.0079 ** 0.0079). (E-G) CD8$^+$ T cells were activated with anti-CD3/CD28. Five hours post activation (T-Early), cells were treated for 3 hr with media containing $^{13}$C-glutamine only, $^{13}$C-glutamine with oligomycin, or $^{13}$C-glutamine with oligomycin plus FCCP. Cell extracts were then subjected for metabolome analysis. (n = 7 biological replicates). (E) Heatmap showing relative amounts of key energy-related metabolites (as indicated in the figure) measured in T-Early cells that were treated with oligomycin or oligomycin plus FCCP. (F) Volcano plot of all analyzed metabolites measured in T-Early cells treated with oligomycin or oligomycin plus FCCP. (G) $^{13}$C-glutamine LC-MS tracing analysis of CD8$^+$ T cells that either were untreated or treated with oligomycin, or oligomycin plus FCCP. Heatmaps summarizing intracellular isotopomers of; Citrate, Glutamate, Glutamine, Succinate, Malate and Aspartate. Statistical method, (A-E) non-parametric Mann–Whitney test, mean ± s.e.m, (F) False Discovery Rate (FDR) P value < 0.05.

The online version of this article includes the following figure supplement(s) for figure 5:

**Figure supplement 1.** Schematic of the suggested model; respiratory restriction through inhibition of complex V by oligomycin leads to increase in mitochondrial membrane potential and decreased electron flow.

## Short exposure to atmospheric oxygen pressure rescues CD8$^+$ T cells' response to lentiviral challenge under systemic hypoxia in vivo

Our findings demonstrated that during early activation, OXPHOS is required primarily to provide ATP for mitochondrial-biogenesis. It is thought that the build-up of additional mitochondrial biomass is a critical checkpoint in T cell activation (*Ron-Harel et al., 2016*; *Buck et al., 2016*). Following the mitochondrial-biogenesis checkpoint, fully activated CD8$^+$ T cells show only a marginal reduction in proliferation capacity under hypoxic conditions (*Doedens et al., 2013*). Thus, our results suggest that during systemic hypoxia, activated CD8$^+$ T cells are arrested at the mitochondrial-biogenesis checkpoint, and might thus be rescued by short oxygen resuscitation.

Building on these insights, we attempted to rescue activated CD8$^+$ T cells that were inhibited by hypoxia by re-exposing them to atmospheric oxygen (*Figure 6A*). Naïve CD8$^+$ T cells were activated under hypoxic or normal atmospheric conditions in vitro. Twenty-four hours later, the CD8$^+$ T cells activated under hypoxia were re-exposed to normal atmospheric conditions or left under hypoxic conditions. As expected, the activated CD8$^+$ T cells that were left under hypoxic conditions for 72 hr remained arrested and showed reduced elevation of CD25 expression compared to control (*Figure 6B-E*). In contrast, CD8$^+$ T cells that were activated under hypoxia and then re-exposed to atmospheric oxygen pressure exhibited significantly increased CD25 expression and proliferative capacity (*Figure 6B-E*).

Next, we tested our hypothesis in vivo (*Figure 6F*). Mice were primed with LV-OVA or left uninfected. Three days after the viral challenge, mice were adoptively transferred with CellTrace-labeled splenocytes from OT1/mito-Dendra2 double transgenic mice. Then the mice were either continuously maintained at atmospheric or 8% oxygen level for 72 hr, or kept at an 8% oxygen level for 24 hr and then transferred to atmospheric oxygen pressure for another 48 hr. In line with the in vitro results, compared to mice kept under systemic hypoxia for 72 hr, the hypoxic mice that were resuscitated at atmospheric oxygen for 48 hr exhibited a significantly improved anti-lentiviral CD8$^+$ T cell response, manifested by a marked increase in proliferative capacity (*Figure 6G–H*). Importantly, resuscitation at atmospheric oxygen pressure also led to increased mtDendra2 expression compared to mice kept under hypoxia for 72 hr (*Figure 6I–J*). These results demonstrate the reversibility of the CD8$^+$ T cell activation arrest mediated by respiratory restriction.

Finally, to examine whether we can utilize our findings to improve anti-viral response in a more clinically relevant approach, we tested whether a short, 24 hr, exposure to atmospheric oxygen pressure would rescue the CD8$^+$ T cell response to Lv-OVA under systemic hypoxia in vivo. We compared markers of CD8$^+$ T cell activation status in mice challenged under atmospheric oxygen pressure, 21% oxygen pressure, continuous systemic hypoxia (8% oxygen pressure), or transient resuscitation (systemic hypoxia followed by 24 hr resuscitation at atmospheric oxygen pressure) (*Figure 6K*). As expected, our analysis of CD62L, CD44, and CD25 revealed that the CD8$^+$ response was strongly inhibited under continuous systemic hypoxia. In contrast, in the transient resuscitation group, we observed a marked increase in the population of OVA-associated CD44$^+$, CD25$^+$, CD62L$^-$ CD8$^+$ T cells (*Figure 6L–Q* and *Figure 6—figure supplement 1A–B*). Importantly the

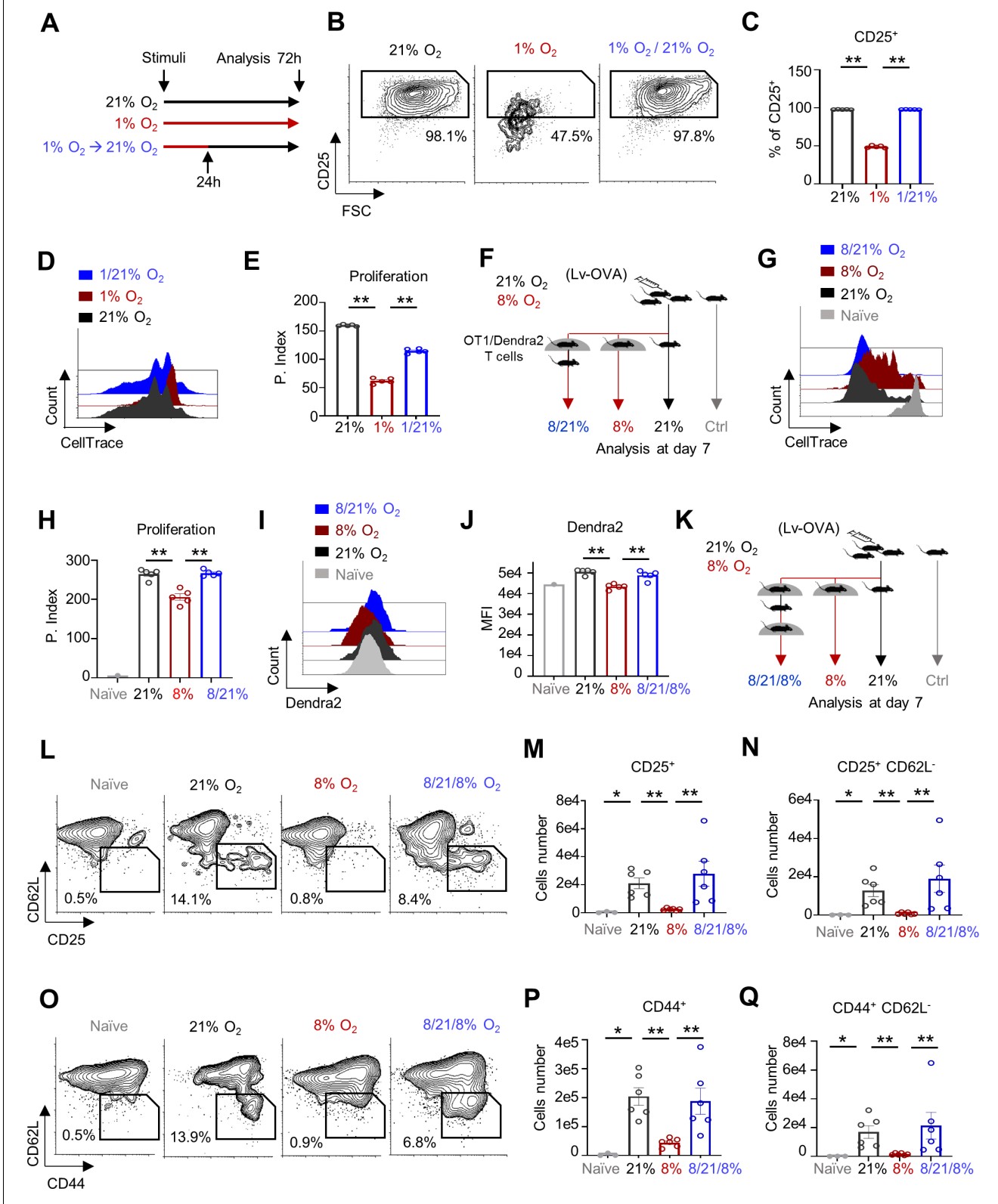

**Figure 6.** Short oxygen exposure rescues CD8+ T cells activated under hypoxia in vivo. (**A**) Schematic describing the experiment in **B-E**. CellTrace-labeled mouse splenocytes were activated with anti-CD3/28 under 1% or atmospheric oxygen pressure (21% O$_2$). Twenty-four hours post activation, a group of cells from the 1% O$_2$ chamber were transferred to 21% O$_2$ (1/21%). The different cell groups were collected at 72 hr post activation and analyzed by flow cytometry. (n = 5 biological replicates). (**B**) Representative flow cytometry plots of FSC vs. CD25 gated on CD8+ T cells. Numbers

*Figure 6 continued on next page*

*Figure 6 continued*

indicate the frequencies of CD25$^+$ cells. (**C**) Bar graph summarizing the results in B. (P value ** 0.0079 ** 0.0079). (**D**) Representative flow cytometry overlay histogram of CellTrace intensity gated on CD8$^+$ T cells from cells that either were grown in 21% O$_2$ (front) or 1% O$_2$ (middle), or from cells that were grown in 1% O$_2$ and transferred to 21% O$_2$ (back). (**E**) Bar graph summarizing the results in D as proliferation index. (P value ** 0.0079 ** 0.0079). (**F**) Schematic of experiment presented in panels **G-J**. C57BL6 mice were primed intradermally in the ear pinna with 5 × 10$^6$ TU of Lv-OVA or left uninfected. Three days after the viral challenge, mice were adoptively transferred i.p. with 4 × 10$^6$ CellTrace-labeled splenocytes from OT1/mito-Dendra2 double transgenic mice. Mice were then divided into four groups; (1) Lv-OVA infected mice grown in 21% O$_2$ (21%) for another 3 days. (2) Lv-OVA infected mice grown in 8% O$_2$ for another 3 days (8%). (3) Lv-OVA infected mice grown in 8% O$_2$ for 24 hr and then transferred to 21% O$_2$ for another 48 hr (8/21%). (4) Lv-OVA uninfected mice grown in 21% O$_2$ (Ctrl). Cells from the deep cervical lymph nodes were then analyzed by flow cytometry analysis. (n = 5 mice for groups 1–3 and n = 3 mice for the control group). (**G**) Representative flow cytometry overlay histogram of CellTrace intensity gated on TCR Vα2$^+$, CD8$^+$, and Dendra$^+$ triple positive cells from each of the four groups. (**H**) Bar graph summarizing the results in **G**, as Proliferation Index. (P value ** 0.0079 ** 0.0079). (**I**) Representative flow cytometry histogram overlay plot of Dendra2 fluorescence intensity gated on the TCR Vα2$^+$, CD8$^+$, and Dendra2$^+$ triple positive cells from each of the four groups. (**J**) Bar graph summarizing the results in I, Dendra2 mean fluorescence intensity (MFI). (P value ** 0.0079 ** 0.0079). (**K**) Schematic of experiment presented in panels (**L-Q**). C57BL6 mice were primed intradermally in the ear pinna with 5 × 10$^6$ TU of Lv-OVA or left uninfected. Twenty-four hours after the viral challenge mice were divided into four groups; (1) Lv-OVA infected mice grown in 21% O$_2$ (21%). (2) Lv-OVA infected mice grown in 8% O$_2$ (8%). (3) Lv-OVA infected mice grown in 8% O$_2$ for 48 hr, transferred to 21% O$_2$ for 24 hr and then transferred back to 8% O$_2$ for an additional 48 hr (8/21/8%). (4) Lv-OVA uninfected mice grown in 21% O$_2$ (Ctrl). Seven days from the beginning of the experiment, cells from the deep cervical lymph nodes were analyzed by flow cytometry. (**L**) Representative flow cytometry plots of CD25 vs. CD62L gated on TCR Vα2$^+$, CD8$^+$ T cells from each of the four groups. Numbers indicate the frequencies of CD25$^+$ CD62L$^-$ cells. (**M**) Bar graph summarizing the number of TCR Vα2$^+$, CD8$^+$, and CD25+ cells in each of the groups. (P value * 0.0238 **0.0022 ** 0.0022). (**N**) Bar graph summarizing the number of TCR Vα2$^+$, CD8$^+$, CD25$^+$ and CD62L$^-$ cells in each of the groups. (P value *0.0238 ** 0.0022 ** 0.0022). (**O**) Same as in L focusing on the CD44$^+$ and CD62L$^-$ cells. (**P**) Bar graph summarizing the number of TCR Vα2$^+$, CD8$^+$, and CD44$^+$ cells in each of the groups. (P value * 0.0238 ** 0.0022 ** 0.0022). (**Q**) Same as in P, focusing on the TCR Vα2$^+$, CD8$^+$, CD44$^+$ and CD62L$^-$ cells. (P value *0.0238 ** 0.0022 ** 0.0022). Statistical method, non-parametric Mann–Whitney test, mean ± s.e.m.

The online version of this article includes the following figure supplement(s) for figure 6:

**Figure supplement 1.** Short oxygen exposure rescues CD8$^+$ T cells activated under hypoxia in vivo.

phenotype in the transient resuscitation group was almost indistinguishable from the control group that was kept under normal atmospheric conditions. Overall, our results demonstrate that the detrimental effect caused by systemic hypoxia in vivo may be alleviated by short exposure to atmospheric oxygen pressure.

## Discussion

Our present results revealed that under systemic chronic hypoxia, CD8$^+$ T cells failed to activate and respond to a viral challenge due to a matrix-localized ATP deficiency that disrupted critical mitochondrial processes.

Several lines of evidence from our study indicated that mitochondrial respiratory-based ATP was not required for T cell cytoplasmic function during activation. Using both genetic and acute models of cytoplasm-specific mitochondrial ATP restriction, we determined that in CD8$^+$ T cells, the ATP demand in the mitochondrial matrix was distinct from that in the cytoplasm. Furthermore, through functional assays, we revealed a matrix-specific ATP crisis following oligomycin mediated acute respiratory-restriction. To confirm these results, we demonstrated that uncoupler-based restimulation of the TCA cycle could functionally rescue respiratory-restricted T-Early cells.

In line with these findings, our metabolic analysis revealed that oligomycin treatment during early activation led to an accumulation of TCA intermediates. Moreover, we demonstrated that addition of the proton uncoupler FCCP substantially reduced the accumulation of mono-phosphonucleotide intermediates, and elevated ATP levels. Since oligomycin maintains ATP synthase arrest, even following the addition of FCCP (*Lee and O'Brien, 2010*), the increase of cellular ATP and reductions of both AMP and GMP may be primarily attributed to matrix-bound substrate-level phosphorylation. Notably, some of these mechanistic observations regarding the inhibitory effect mediated by an acute respiratory restriction on CD8$^+$ T cell activation, were based on the application of oligomycin. Since oligomycin, only partially mimics hypoxia, follow-up work should look into further mechanistic effects induced by hypoxia.

Taken together, our results suggested that under chronic hypoxia, activated CD8$^+$ T cells were arrested at the mitochondrial-biogenesis checkpoint, and could be rescued by oxygen resuscitation.

Building on these insights, we demonstrated that hypoxia-arrested CD8$^+$ T cells in vivo could be rescued by short exposure to atmospheric conditions.

Overall, our present study revealed that hypoxia had detrimental effects on mitochondrial-biogenesis in activated CD8$^+$ T cells suggest a potential new approach to the reduction of viral infections in hypoxia-associated diseases.

# Materials and methods

## Key resources table

| Reagent type (species) or resource | Designation | Source or reference | Identifiers | Additional information |
|---|---|---|---|---|
| Antibody | Anti-mouse-CD8α (Rat Monoclonal) clone 53–6.7 | Biolegend | Cat# 10071 Cat# 10072 Cat# 10072 | FACS 1:500 |
| Antibody | Anti-mouse-CD44 (Rat Monoclonal) clone IM7 | Biolegend | Cat# 10452 | FACS 1:1000 |
| Antibody | Anti-mouse-CD69 (Armenian Hamster Monoclonal) clone H1.2F3 | Biolegend | Cat# 10451 | FACS 1:500 |
| Antibody | Anti-mouse-CD25 (Rat Monoclonal) clone 3C7 | Biolegend | Cat# 10190 | FACS 1:500 |
| Antibody | Anti-mouse-CD62L (Rat Monoclonal) clone MEL-14 | Biolegend | Cat# 10441 | FACS 1:500 |
| Antibody | Anti-mouse TCR Vα2 (Rat Monoclonal) clone B20.1 | Biolegend | Cat# 12780 | FACS 1:1000 |
| Antibody | Anti-mouse-Ki67 (Rat Monoclonal) clone 16A8 | Biolegend | Cat# 652423 | FACS 1:100 |
| Antibody | Purified anti mouse-C3ε (Armenian Hamster monoclonal) clone 145–2 C11 | Biolegend | Cat# 100340 | Activation 0.1 µg/ml |
| Antibody | Purified anti mouse-CD28 (Syrian Hamster monoclonal) clone 37.51 | Biolegend | Cat# 102116 | Activation 0.1 µg/ml |
| Antibody | Purified anti-human-CD3ε (Mouse monoclonal) clone OKT3 | Biolegend | Cat# 317326 | Activation 0.1 µg /ml |
| Antibody | Purified anti-human-CD28 (Mouse monoclonal) clone CD28.2 | Biolegend | Cat# 302934 | Activation 0.1 µg/ml |
| Antibody | Anti-human-CD8α (Mouse monoclonal) clone HIT8a | Biolegend | Cat# 30090 | FACS 1:400 |

*Continued on next page*

*Continued*

| Reagent type (species) or resource | Designation | Source or reference | Identifiers | Additional information |
|---|---|---|---|---|
| Antibody | Anti-human-CD25 (Mouse monoclonal) clone M-A251 | Biolegend | Cat# 35610 | FACS 1:500 |
| Antibody | Anti-mouse AMPKα (Rabbit monoclonal) | Cell Signalling | Cat# 2532 | WB 1:1000 |
| Antibody | Anti-mouse phospho-AMPKα (Rabbit monoclonal) | Cell Signalling | Cat#: 2531 | WB 1:1000 |
| Antibody | Donkey Anti-Rabbit IgG H and L (HRP) (Donkey polyclonal) | abcam | Cat# ab97085 | WB 1:10000 |
| Antibody | Anti-Ubiquitin (Mouse monoclonal) clone FK2 | Merck-Millipore | Cat# ST1200 | IP 2 µg |
| Chemical compound, drug | Oligomycin A | Cayman Chemicals | Cat# 11342 | 1 nM - 1 µM |
| Chemical compound, drug | Rotenone | Cayman Chemicals | Cat# 13995 | 1 µM |
| Chemical compound, drug | Antimycin A | Cayman Chemicals | Cat# 19433 | 1 µM |
| Chemical compound, drug | FCCP | Cayman Chemicals | Cat# 15218 | 1 µM |
| Chemical compound, drug | Bongkrekic Acid (ammonium salt) | Cayman Chemicals | Cat# 19079 | 1 µM - 2 µM |
| Chemical compound, drug | EZview Red Protein G Affinity Gel | Sigma-Aldrich | Cat# E3403 | |
| Chemical compound, drug | Protease Inhibitor Cocktail | Sigma-Aldrich Israel | P8340 | WB and IP 1:100 |
| Sequence-based reagent | *Ubc* F | This paper | PCR primers | GCCCAGTGTTACCACCAAGA |
| Sequence-based reagent | *Ubc* R | This paper | PCR primers | CCCATCACACCCAAGAACA |
| Sequence-based reagent | *Rpl13* F | This paper | PCR primers | ATGACAAGAAAAAGCGGATG |
| Sequence-based reagent | *Rpl13* R | This paper | PCR primers | CTTTCCTGCCTGTTTCCGTA |
| Sequence-based reagent | *mt-Rnr* F | This paper | PCR primers | CATACTGGAAAGTGTGCTTGGA |
| Sequence-based reagent | *mt-Rnr* R | This paper | PCR primers | GTGTAGGGCTAGGGCTAGGA |

*Continued on next page*

*Continued*

| Reagent type (species) or resource | Designation | Source or reference | Identifiers | Additional information |
|---|---|---|---|---|
| Sequence-based reagent | *mt-Rnr1* F | This paper | PCR primers | ACCGCGGTC ATACGATTAAC |
| Sequence-based reagent | *mt-Rnr1* R | This paper | PCR primers | CCCAGTTTGG GTCTTAGCTG |
| Sequence-based reagent | *mt-Rnr2* F | This paper | PCR primers | GGGATAACAGC GCAATCCTA |
| Sequence-based reagent | *mt-Rnr2* R | This paper | PCR primers | GATTGCTCCG GTCTGAACTC |
| Commercial assay, kit | MitoProbe TMRM Assay Kit for Flow Cytometry | Thermo Fischer: | Cat# M20036 | FACS 50 nM |
| Commercial assay, kit | ProteaseMAX Surfactant | Promega Corp | Cat# V2071 | |
| Commercial assay, kit | CellTrace Violet Cell Proliferation Kit, for flow cytometry | Thermo Fischer: Molecular Probes | Cat# C34571 | FACS 1:100 |
| Commercial assay, kit | EasySep Mouse CD8$^+$ T Cell Isolation Kit | STEMCELL Technologies | Cat# 19853A | |
| Commercial assay, kit | Direct-zol RNA MiniPrep Plus | Zymo Research | Cat# R2071 | |
| Commercial assay, kit | ProtoScript First Strand cDNA Synthesis Kit | New England BioLabs, Inc | Cat# E6300L | |
| Commercial assay, kit | Power SYBR Green PCR Master Mix | Applied Biosystems | Cat# 4367660 | |
| Strain, strain background *Mus musculus* | C57BL/6J | Jackson Laboratory | Stock No: 000664 | Wild type |
| Strain, strain background *Mus musculus* | Slc25a5tm 1.1Nte/J | Jackson Laboratory | Stock No: 029482 | ANT2flox/lox |
| Strain, strain background *Mus musculus* | C57BL/6-Tg(TcraTcrb) 1100Mjb/J | Jackson Laboratory | Stock No: 003831 | OT1 |
| Strain, strain background *Mus musculus* | B6.Cg-Tg (Lck-cre) 1CwiN9 (Lck-Cre) | Taconic | Model # 4197 | Lck-Cre |
| Strain, strain background *Mus musculus* | Gt(ROSA) 26Sortm1.1(CAG-Mito-Dendra2) Dcc | Dr. Tsvee Lapidot from the Weizmann Institute of Science | | mito-Dendra2 |
| Recombinant DNA reagent | Lv-OVA-GFP | Dr. Avihai Hovav from the Hebrew University of Jerusalem | | Ovalbumin and GFP expressing lentiviral plasmid |
| Recombinant DNA reagent | pCMV-VSV-G | a gift from Bob Weinberg https://www. addgene.org/8454/ | Addgene Plasmid #8454 | VSV-G envelope expressing plasmid |

*Continued on next page*

*Continued*

| Reagent type (species) or resource | Designation | Source or reference | Identifiers | Additional information |
|---|---|---|---|---|
| Recombinant DNA reagent | psPAX2 | a gift from Didier Trono https://www.addgene.org/12260/ | Addgene Plasmid #12260 | Lentiviral packaging plasmid |
| Software, algorithm | Kaluza software | Beckman Coulter | | FACS acquisition software |
| Software, algorithm | FACS Express 6 | De Novo Software | | FACS analysis software |
| Software, algorithm | Seahorse Wave | Agilent | | OCR and EACAR analysis |
| Software, algorithm | Perseus | | | Metabolic and proteomic analysis |
| Software, algorithm | Prism 8 | GraphPad | | Graphs and Heatmaps, statistical analysis |
| Software, algorithm | Thermo Xcalibur | Thermo Fisher Scientific | | Metabolomics LC-MS data acquisition |
| Software, algorithm | TraceFinder 4.1 | Thermo Fisher Scientific | | Metabolomics LC-MS data analysis |

## Mice

The C57BL/6J (wild-type), Slc25a5tm1.1Nte/J (*ANT2^flox/lox^*), and C57BL/6-Tg(TcraTcrb)1100Mjb/J (OT1) mice were from The Jackson Laboratory. B6.Cg-Tg(Lck-cre)1CwiN9 (Lck-Cre) were from Taconic. The T cell-specific ANT2 knockout mice were generated by crossing mice containing a conditional floxed allele of ANT2 Slc25a5tm1.1Nte/J (*ANT2flox/lox*) with transgenic mice expressing Cre under the control of the Lck gene promoter (Lck-Cre). Gt(ROSA)26Sortm1.1(CAG-Mito-Dendra2) Dcc (mito-Dendra2) mice were a kind gift from Dr. Tsvee Lapidot from the Weizmann Institute of Science. Mice were maintained and bred under specific pathogen free conditions in the Hebrew University animal facilities according to Institutional Animal Care and Use Committee regulations. All mice were maintained on the C57BL/6J background and used for experiments at 8–12 weeks of age.

## Quantitative real-time PCR and cDNA preparation

Total RNA from purified CD8^+ T cells was extracted with Direct-zol RNA MiniPrep Plus (Zymo Research) following DNA removal step. cDNA was synthesized using ProtoScript First Strand cDNA Synthesis Kit (New England BioLabs, Inc – E6300L) with random primers for the MT-RNR transcripts, and oligo-dT primers for all other transcripts. Quantitative real-time PCR was then performed using Applied Biosystems (AB), Viia 7 Real-Time PCR system with a Power SYBR green PCR master mix kit (Applied Biosystems).

Reaction was performed as follow:

i. 50℃ 2 min, one cycle
ii. 95℃ 10 min, one cycle
iii. 95℃ 15 s - > 60℃ 1 min, 40 cycles
iv. 95℃ 15 s, one cycle
v. 60℃ 1 min, one cycle
vi. 95℃ 15 s, one cycle

Data was normalized to Mouse endogenous control (UBC and or RPL13) and analyzed using ΔΔCt model unless else is indicated.

Each experiment was performed in sixplicates and was repeated three times. Student's t-test was used with 95% confidence interval.

## Primers used for quantitative Real-Time PCR

| Gene | Forward | Reverse |
|------|---------|---------|
| Ubc | GCCCAGTGTTACCACCAAGA | CCCATCACACCCAAGAACA |
| Rpl13 | ATGACAAGAAAAAGCGGATG | CTTTCCTGCCTGTTTCCGTA |
| mt-Rnr | CATACTGGAAAGTGTGCTTGGA | GTGTAGGGCTAGGGCTAGGA |
| mt-Rnr1 | ACCGCGGTCATACGATTAAC | CCCAGTTTGGGTCTTAGCTG |
| mt-Rnr2 | GGGATAACAGCGCAATCCTA | GATTGCTCCGGTCTGAACTC |

## Protein mass spectrometry

### Sample preparation

Agarose beads containing immunoprecipitated samples, frozen at −20°C was subject to tryptic digestion, performed in the presence of 0.05% ProteaseMAX Surfactant (from Promega Corp., Madison, WI, USA). The peptides were then desalted on C18 Stage tips (*Rappsilber et al., 2007*). A total of 0.5 µg of peptides were injected into the mass spectrometer.

### LC-MS/MS analysis

MS analysis was performed using a Q Exactive Plus mass spectrometer (Thermo Fisher Scientific) coupled on-line to a nanoflow UHPLC instrument (Ultimate 3000 Dionex, Thermo Fisher Scientific). Eluted peptides were separated over a 60 min gradient run at a flow rate of 0.3 µl/min on a reverse phase 25-cm-long C18 column (75 µm ID, 2 µm, 100 Å, Thermo PepMapRSLC). The survey scans (380–2,000 m/z, target value 3E6 charges, maximum ion injection times 50 ms) were acquired and followed by higher energy collisional dissociation (HCD) based fragmentation (normalized collision energy 285). A resolution of 70,000 was used for survey scans and up to 15 dynamically chosen most abundant precursor ions were fragmented (isolation window 1.6 m/z). The MS/MS scans were acquired at a resolution of 17,500 (target value 5E4 charges, maximum ion injection times 57 ms). Dynamic exclusion was 60 s.

### MS data analysis

Mass spectra data were processed using the MaxQuant computational platform, version 1.5.3.12. Peak lists were searched against the *Homo sapiens* Uniprot FASTA sequence database containing a total of 26,199 reviewed entries or a custom FATSA file containing mouse mitochondrial leader peptides. The search included cysteine carbamidomethylation as a fixed modification and oxidation of methionine as variable modifications. Peptides with minimum of seven amino-acid length were considered and the required FDR was set to 1% at the peptide and protein level. Protein identification required at least three unique or razor peptides per protein group. The dependent-peptide and match-between-runs options were used.

## In vivo viral challenge under chronic hypoxia

C57BL6 mice were primed intradermally in the ear pinna with $5 \times 10^6$ transduction units (TU) of Lv-OVA or left untreated. Twenty-four hours following the viral challenge, mice were transferred to chambers for additional 6 days and kept under either 8% or 21% oxygen pressure. Extracted cells from the deep cervical lymph nodes were then analyzed by flow cytometry as follows.

## In vivo T cell proliferation assay

C57BL6 mice were primed intradermally in the ear pinna with $5 \times 10^6$ TU of Lv-OVA. Three days after the viral challenge, mice were adoptively transferred i.p. with $4 \times 10^6$ CellTrace-labeled splenocytes from OT1/mito-Dendra2 double transgenic mice. Three days later cells from the deep cervical lymph nodes were analyzed by flow cytometry analysis.

### In vitro T cell proliferation assay

Splenocytes or human PBMCs were stained with CellTrace (Molecular Probes, Eugene, OR) prior to activation for 30 min at 37°C. Cells were then activated in 24-flat-well plates ($5 \times 10^6$ cells per well) or 96-flat-well plates ($1 \times 10^6$ cells per well) with soluble anti-CD3ε (1 µg/ml) and anti-CD28 (1 µg/ml). Proliferation index reflects the sum the percentage of cells in each generation group multiplied by the number of division.

### Antibodies

The following antibodies were used for flow cytometry: anti-CD8α (53–6.7), anti-CD44 (IM7), anti-CD69 (H1.2F3), anti-CD25 (3C7), anti-CD62L (MEL-14), anti-mouse TCR Vα2 (B20.1), anti-human CD8 (HIT8a), anti-human CD25 (M-A251), and anti-mouse Ki67 (16A8). All antibodies were from BioLegend.

Purified anti-CD3ε (145–2C11) and anti-CD28 (37.51; both from Biolegend) were used at the appropriate concentration for mouse T cell activation. Purified anti-CD3ε (OKT3) and anti-CD28 (CD28.2; both from Biolegend) were used at the appropriate concentration for human T cell activation.

Antibody to AMPKα phosphorylated on Thr172 and anti-AMPKα both from Cell Signaling Technology were used for immunoblot analysis.

Antibody to Ubiquitinated proteins (FK2) from Merck-Millipore was used for the immunoprecipitation assay.

### Flow cytometry

Cells were stained with various conjugated mAbs against cell-surface markers in FACS buffer (PBS containing 1% FBS and 1 mM EDTA) for 30 min at 4°C. For mitochondrial membrane potential staining, cells were labeled with TMRM 50 nM (Molecular Probes, Eugene, OR) in FACS buffer without EDTA for 30 min at 30°C. Stained cells were analyzed by Gallios flow cytometer with Kaluza software (Beckman Coulter, Brea, CA) and analyzed by FACS Express 6 (De Novo Software).

### Metabolism assays

OCR and ECAR were measured using a 24-well XF extracellular flux analyzer (EFA) (Seahorse Bioscience). Purified naive or activated CD8$^+$ T cells ($1 \times 10^6$ cells per well) were seeded in Seahorse XF24 designated plates using Cell-Tak (Corning) adherent and assayed according to manufacturer instructions.

### Western blot and immunoprecipitation

Purified naïve or activated CD8$^+$ T cells were lysed in radioimmunoprecipitation assay (RIPA) buffer; 10 µg protein from each sample was separated by SDS–PAGE, and immunoblotted with anti AMPKα antibody or p-AMPKα (Thr172) (Cell Signaling, Danvers, MA; 2532) followed by peroxidase donkey anti-rabbit IgG (Jackson Laboratory; 711-005-152).

For immunoprecipitation extracts from purified activated CD8$^+$ T cells were prepared in extraction buffer (50 mM Tris-HCl, pH 8.0, 5 mm EDTA, 150 mM NaCl and 0.5% NP-40, supplemented with Protease Inhibitor Cocktail, Sigma-Aldrich, Israel). Protein extracts were then precleared with protein G beads (EZview Red Protein G Affinity Gel, Sigma-Aldrich, Israel), following incubation for 30 min at 4°C. Protein G beads were pelleted out, and the supernatant was taken for immunoprecipitation with 2 µg of anti-ubiquitin antibody (FK2, Merck-Millipore) for 12 hr at 4°C. Immune complexes were pelleted with protein G beads as before, and the pellets were washed three times in buffer B (5% sucrose, 50 mM Tris-HCl pH 7.4, 500 mM NaCl, 5 mM EDTA and 0.5% NP-40), followed by three washes with buffer C (50 mM Tris-HCl pH 7.4, 150 mM NaCl and 5 mM EDTA). The precipitated proteins were then subjected to MS analysis.

### Targeted metabolic analysis

CD8$^+$ T cells were cultured in either anti-CD3/CD28 coated or uncoated 96 well plate (1 million cells/well), suspended in RPMI supplemented with 10% dialyzed Fetal Bovine Serum and 100 µM Alanine with or without labeled glutamine. Following 5 or 24 hr activated cells were treated with 500

nM Oligomycin, Oligomycin and 1 µM FCCP or left untreated for 2 hr. Naïve, and activated cells were then extracted for metabolomics LC-MS analysis.

## Medium extracts

Twenty microliters of culture medium was added to 980 µl of a cold extraction solution (−20°C) composed of methanol, acetonitrile, and water (5:3:2). Cell extracts: Cells were rapidly washed three times with ice-cold PBS, after which intracellular metabolites were extracted with 100 µl of ice-cold extraction solution for 5 min at 4°C. Medium and cell extracts were centrifuged (10 min at 16,000 g) to remove insoluble material, and the supernatant was collected for LC-MS analysis. Metabolomics data was normalized to protein concentrations using a modified Lowry protein assay.

LC-MS metabolomics analysis was performed as described previously (*Mackay et al., 2015*). Briefly, Thermo Ultimate 3000 high-performance liquid chromatography (HPLC) system coupled to Q- Exactive Orbitrap Mass Spectrometer (Thermo Fisher Scientific) was used with a resolution of 35,000 at 200 mass/charge ratio (m/z), electrospray ionization, and polarity switching mode to enable both positive and negative ions across a mass range of 67 to 1000 m/z. HPLC setup consisted ZIC-pHILIC column (SeQuant; 150 mm x 2.1 mm, 5 µm; Merck), with a ZIC-pHILIC guard column (SeQuant; 20 mm x 2.1 mm). 5 µl of Biological extracts were injected and the compounds were separated with mobile phase gradient of 15 min, starting at 20% aqueous (20 mM ammonium carbonate adjusted to pH.2 with 0.1% of 25% ammonium hydroxide) and 80% organic (acetonitrile) and terminated with 20% acetonitrile. Flow rate and column temperature were maintained at 0.2 ml/min and 45°C, respectively, for a total run time of 27 min. All metabolites were detected using mass accuracy below five ppm. Thermo Xcalibur was used for data acquisition. TraceFinder 4.1 was used for analysis. Peak areas of metabolites were determined by using the exact mass of the singly charged ions. The retention time of metabolites was predetermined on the pHILIC column by analyzing an in-house mass spectrometry metabolite library that was built by running commercially available standards.

## Statistical analysis

The statistical significance of differences was determined by the two-tailed Mann-Whitney non-parametric *t*-test. Biological replicates refer to independent experimental replicates sourced from different mice/human donors. Technical replicates refer to independent experimental replicates from the same biological source. Differences with a *P* value of less than 0.05 were considered statistically significant. Graph prism and Perseus programs were used. MS data was normalized by ranking, when applicable, non-values were plugged with replicates mean to prevent zeros bias.

## Additional information

### Funding

| Funder | Grant reference number | Author |
| --- | --- | --- |
| Israeli Science Foundation | Personal grant | Michael Berger |
| German-Israeli Foundation for Scientific Research and Development | I-1474-414.13/2018 | Michael Berger |
| Israeli Science Foundation | 1596/17 | Michael Berger |

The funders had no role in study design, data collection and interpretation, or the decision to submit the work for publication.

### Author contributions

Amijai Saragovi, Conceptualization, Data curation, Formal analysis, Investigation, Methodology, Writing - original draft, Project administration, Writing - review and editing; Ifat Abramovich, Ibrahim Omar, Eliran Arbib, Investigation; Ori Toker, Resources; Eyal Gottlieb, Conceptualization; Michael Berger, Conceptualization, Supervision, Funding acquisition, Investigation, Methodology, Writing - original draft, Writing - review and editing

### Author ORCIDs
Amijai Saragovi ORCID https://orcid.org/0000-0002-4376-1390
Michael Berger ORCID https://orcid.org/0000-0002-3469-0076

### Ethics
Human subjects: Human blood samples were obtained via Shaare Zedek Medical Center Jerusalem, Helsinki committee approval number: 143/14.
TAnimal experimentation: This study was performed in strict accordance with the guidelines of the institutional ethics committee (AAALAC standard). The protocols were approved by the Committee on the Ethics of Animal Experiments of the Hebrew University (Ethics Committee - research number: MD-16-14863-1 and MD-18-15662-5). Every effort was made to minimize suffering.

### Decision letter and Author response
Decision letter https://doi.org/10.7554/eLife.56612.sa1
Author response https://doi.org/10.7554/eLife.56612.sa2

## Additional files

### Supplementary files
• Supplementary file 1. Matrix protein annotated by GO with either ATP or GTP domains.

• Transparent reporting form

### Data availability
Metabolic analysis data and Protein MS analysis have been deposited in OSF under https://doi.org/10.17605/OSF.IO/JKMQF.

The following dataset was generated:

| Author(s) | Year | Dataset title | Dataset URL | Database and Identifier |
|---|---|---|---|---|
| Amijai S, Ifat A, Ibrahim O, Eliran A, Ori T, Eyal G, Michael B | 2020 | MS analysis of anti-ubiquitin precipitated proteins from oligomycin treated T cells | https://doi.org/10.17605/OSF.IO/JKMQF | Open Science Framework, 10.17605/OSF.IO/JKMQF |

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
