## [Decision Letter]

**Acceptance summary:**

In this study, Saragovi and colleagues demonstrate that mitochondrial respiration is critical during these first hours to enable TCA cycle activity and produce ATP within mitochondrial matrix to support mitochondrial-biogenesis. The authors show a detrimental effect of hypoxia/oligomycin on mitochondrial-biogenesis during T cell activation. They find that ATP-dependent matrix processes, critical for mitochondrial-biogenesis, are impaired under respiratory restriction condition, leading to compromised T cell activation. Moreover, increased mitochondrial matrix-localized ATP via boosting substrate-level phosphorylation can partially rescue the defect upon respiratory restriction. Thus, they refute a common hypothesis that mitochondrial activity is critical for providing ATP to the cytosol, since inhibition of ATP export does not inhibit T cell activation. Lastly, they show that anti-viral responses under systemic oxygen restriction in vivo can be rescued by short exposure to atmospheric oxygen pressure. Thus, the authors provide interesting evidence regarding linkages between hypoxia, mitobiogenesis and T cell activation.

**Decision letter after peer review:**

Thank you for submitting your article "Systemic hypoxia inhibits T cell response by limiting mitobiogenesis via matrix substrate-level phosphorylation arrest" for consideration by *eLife*. Your article has been reviewed by two peer reviewers, and the evaluation has been overseen by a Reviewing Editor and Satyajit Rath as the Senior Editor. The following individual involved in review of your submission has agreed to reveal their identity: Noga Ron-Harel (Reviewer #2).

The reviewers have discussed the reviews with one another and the Reviewing Editor has drafted this decision to help you prepare a revised submission.

As the editors have judged that your manuscript is of interest, but as described below that additional experiments are required before it is published, we would like to draw your attention to changes in our revision policy that we have made in response to COVID-19 (https://elifesciences.org/articles/57162). First, because many researchers have temporarily lost access to the labs, we will give authors as much time as they need to submit revised manuscripts. We are also offering, if you choose, to post the manuscript to bioRxiv (if it is not already there) along with this decision letter and a formal designation that the manuscript is 'in revision at *eLife*'. Please let us know if you would like to pursue this option. (If your work is more suitable for medRxiv, you will need to post the preprint yourself, as the mechanisms for us to do so are still in development.)

Summary:

Early T cell activation initiates a robust program of mitochondrial-biogenesis and metabolic rewiring, and induces mitochondrial respiration. While previous studies have suggested that there is a critical time window of mitochondrial activity for T cell activation, the mechanistic basis of this dependency has been unclear. In this study, Saragovi and colleagues demonstrate that mitochondrial respiration is critical during these first hours to enable TCA cycle activity and produce ATP within mitochondrial matrix to support mitochondrial-biogenesis. The authors show a detrimental effect of hypoxia on mitochondrial-biogenesis during T cell activation. They use oligomycin treatment to mimic respiratory restriction and find that ATP-dependent matrix processes, critical for mitochondrial-biogenesis, are impaired under respiratory restriction condition, leading to compromised T cell activation. Moreover, increased mitochondrial matrix-localized ATP via boosting substrate-level phosphorylation can partially rescue the defect upon respiratory restriction. Thus, they refute a common hypothesis that mitochondrial activity is critical for providing ATP to the cytosol, since inhibition of ATP export does not inhibit T cell activation. Lastly, they show that anti-viral responses under systemic oxygen restriction in vivo can be rescued by short exposure to atmospheric oxygen pressure. Thus, the authors provide convincing evidence to demonstrate the linkages between hypoxia, mitobiogenesis and T cell activation.

However, a number of major concerns, identified below, remain to be addressed in order to increase enthusiasm for publication.

Essential revisions:

1) The authors use oligomycin treatment to establish respiratory restriction and mimic a hypoxic environment. However, it is unclear whether the impaired T cell activation coupled with ATP-deficiency and blocked mitobiogenesis upon oligomycin treatment is similar to the phenotypes observed under hypoxic conditions. It is therefore essential to examine if hypoxia also impairs T cell activation when encountered at early but not at later stages, and whether the mitobiogenesis and ATP-dependent matrix processes are also inhibited under hypoxic conditions.

Similarly, it is advisable to examine if oligomycin-mediated effects can also be reversed as hypoxic effects can be (Figure 6A-6D).

2) While the model chosen for in vivo studies is appropriate (Figures 1 and 6), the data provided require additions of further analysis of T cell activation and effector functions under the different experimental conditions used.

In Figure 1 and in Figure 6, which essentially repeats the experiment in Figure 1 and tests whether a short exposure to high oxygen can rescue T cell activation under hypoxia, the authors show reduction in CD62L expression as the only parameter of T cell activation. It will be more convincing to include additional activation markers (such as CD25, CD69), especially since those are used in other experiments presented in the paper, and the sporadic usage of different markers is confusing. Assays for T cell proliferation (using Ki67, for example) and effector functions (IFNγ, TNFa, GzB levels) under hypoxia are also advisable.

In the experiments for Figure 6 day 3 was chosen for normoxia; but using an earlier time point would be more congruent with the ~9 h in-vitro time window.

Finally, it would be important to test if mitobiogenesis is also rescued after short oxygen exposure, so as to further identify a specific time window for mitobiogenesis essential for T cell activation.

3) In order to enhance the observations on AMPK signaling (Figure 3), it would be useful to include treatment with AMPK inhibitors in order to conclude that activated AMPK signaling is independent of the impaired activation upon respiratory restriction.

4) While investigating the role of ATP transportation, it would be best to examine the activation phenotypes in ANT2-depleted or Bongkrekic acid-treated T-Early and T-Late T cells upon oligomycin treatment or hypoxia conditions Figures 3D-G).

5) The impaired cellular processes observed in T-Early, not in T-Late, T cells (Figure 4) could indicate either that mitochondrial-biogenesis is ongoing and essential for the early phase of T cell activation and is less critical in T-Late cells, or that these cellular processes in T-Late cells are independent of respiratory restriction. Data comparing these cellular processes in untreated T-Early and T-Late cells to test if mitobiogenesis has already declined in T-Late cells in comparison with T-Early cells would be useful to distinguish between these. Thus, for example, in Figure 4C, all groups could be normalised to the untreated 9 h group.

Similarly, with respect to the importance of mitobiogenesis in the development of tolerance, it would be useful to examine mitobiogenesis or mitochondrial mass in T-Early and T-Late cells in Figure 4A.

6) In the examination of cell size, CD25 expression and proliferation in oligomycin-treated T cells (Figure 2A), analyses of percentages of CD44- and CD25-positive cells and the expression levels of the activation markers in activated T cells would be very useful.

7) In Figure 6, there appears to be some confusion that needs clarification. The bar graph in Figure 6F shows higher levels of effector T cells (CD62L^-^) in mice kept under hypoxia, which is the opposite of what the authors claim. In Figure 6G, showing CD62L expression in antigen-specific CD8 T cells, the authors claim that the data are showing reduced CD62L expression in mice exposed to high oxygen. However, the difference in the presented histograms seem to be in the number of cells acquired.

8) The language in the manuscript is very difficult to follow, and revision would be very helpful. Careful editing is also needed; an example is of panels that are mentioned in the text but are missing from the figures (S3B-S3G).

[Editors' note: further revisions were suggested prior to acceptance, as described below.]

Thank you for submitting your article "Systemic hypoxia inhibits T cell response by limiting mitobiogenesis via matrix substrate-level phosphorylation arrest" for consideration by *eLife*. Your article has been reviewed by two peer reviewers, and the evaluation has been overseen by a Reviewing Editor and Satyajit Rath as the Senior Editor. The following individual involved in review of your submission has agreed to reveal their identity: Noga Ron-Harel (Reviewer #2).

The reviewers have discussed the reviews with one another and the Reviewing Editor has drafted this decision to help you prepare a revised submission.

Summary:

Early T cell activation initiates a robust program of mitochondrial-biogenesis and metabolic rewiring, and induces mitochondrial respiration. While previous studies have suggested that there is a critical time window of mitochondrial activity for T cell activation, the mechanistic basis of this dependency has been unclear. In this study, Saragovi and colleagues demonstrate that mitochondrial respiration is critical during these first hours to enable TCA cycle activity and produce ATP within mitochondrial matrix to support mitochondrial-biogenesis. The authors show a detrimental effect of hypoxia/oligomycin on mitochondrial-biogenesis during T cell activation. They use oligomycin treatment to mimic respiratory restriction and find that ATP-dependent matrix processes, critical for mitochondrial-biogenesis, are impaired under respiratory restriction condition, leading to compromised T cell activation. Moreover, increased mitochondrial matrix-localized ATP via boosting substrate-level phosphorylation can partially rescue the defect upon respiratory restriction. Thus, they refute a common hypothesis that mitochondrial activity is critical for providing ATP to the cytosol, since inhibition of ATP export does not inhibit T cell activation. Lastly, they show that anti-viral responses under systemic oxygen restriction in vivo can be rescued by short exposure to atmospheric oxygen pressure. Thus, the authors provide interesting evidence regarding linkages between hypoxia, mitobiogenesis and T cell activation. A number of major concerns identified have been mostly addressed in the revised manuscript. However, a couple of issues still remain as identified below.

1) The use of oligomycin to mimic hypoxia remains a problem, since it does not simply mimic hypoxia. If it can be achieved, it is very advisable to perform these experiments with hypoxia. At the very least, the conclusions of this manuscript should be toned down substantially in the interim.

2) The cell death of early activated naive T cells induced by compound C may be a dosage-related artificial result. It is not clear whether the authors examine different doses. Some controls should be included to make the conclusion made. At the very least, the authors need to tone done their conclusions of these data in the interim.

Revisions expected in follow-up work:

If the additional oligomycin- and compound C-related work mentioned above cannot be provided now, data addressing these concerns would be expected in follow-up work.

---

## [Author Response]

Essential revisions:1) The authors use oligomycin treatment to establish respiratory restriction and mimic a hypoxic environment. However, it is unclear whether the impaired T cell activation coupled with ATP-deficiency and blocked mitobiogenesis upon oligomycin treatment is similar to the phenotypes observed under hypoxic conditions. It is therefore essential to examine if hypoxia also impairs T cell activation when encountered at early but not at later stages, and whether the mitobiogenesis and ATP-dependent matrix processes are also inhibited under hypoxic conditions.

Oligomycin partially mimics the effect induced by hypoxia by imposing cellular respiratory restriction (Chang et al., 2013) (Solaini et al., 2010) (Sgarbi et al., 2018). We chose to use oligomycin for two main reasons: 1. It provides a simple experimental system to test the immediate effects of respiratory restriction under multiple conditions. 2. as opposed to hypoxia, it allows to distinguish between the indirect effects mediated by the inhibition of the electron transport chain and the TCA cycle (Martínez-Reyes et al., 2016), and the direct effect, as a result of reduced mitochondrial ATP (Lee and O’Brien, 2010). It is worthy to mention that there are several papers that already showed that hypoxia leads to impaired activation of naïve T cells in vitro, but none of them provided mechanistic insight that will explain why oxygen is necessary for activation of naïve T cells. Applying oligomycin and later combination of oligomycin and FCCP allows a fine dissection of the direct role of oxygen in T cell activation. Hypoxia is an evolving process that leads to accumulative effects. When the experimental conditions were suitable, we repeated the experiment using a hypoxic chamber. However, due to multiple reasons such as oxygen pressure in the media, the hypoxic chamber method is not rapid enough to meet the requirements in some of the more mechanistic experiments. For example testing the immediate-direct influence of ATP-dependent matrix processes requires precise timing. Please note that these sets of experiments are performed by applying oligomycin to a very short period of ~1h. This cannot be done by applying hypoxic conditions.

Following the reviewers' comments we now provide the additional data as follow:

New Figure 2A-E: CD8^+^ T cells were activated in vitro and subjected to hypoxia at early and late time points post stimuli. Indeed, CD8^+^ T cells that were exposed to hypoxia at an early time point (5h) following activation demonstrated an impaired elevation in CD25 expression and decreased proliferative capacity. These impairments were shown to be significantly improved in cells that were transferred to hypoxic conditions at a late time point (18h) following activation (Figure 2B-E). These results suggest that, similar to oligomycin-treatment, hypoxia impairs T cell activation when encountered at early but not at later stages.

New Figure 4A-D: using CD8^+^ T cells from mito-Dendra2 we show that the intensity of Dendra2 in activated CD8^+^ T cells failed to be elevated when the cells were exposed to 1% oxygen 5 hours but not 18 hours following activation. These results demonstrate that, similar to oligomycin treatment, mitochondrial-biogenesis is inhibited under hypoxic conditions.

Similarly, it is advisable to examine if oligomycin-mediated effects can also be reversed as hypoxic effects can be (Figure 6A-6D).

Oligomycin is an irreversible ATP synthase inhibitor (Matsuno-Yagi A, Hatefi Y. J Biol Chem. 1993;268(3):1539-45. PMID: 8380571). Therefore the effects of oligomycin cannot be reversed by simply washing it from the medium. However, we demonstrated that the effect on T cell activation mediated by oligomycin can be rescued by FCCP, suggesting that the effect of oligomycin on cellular respiration may be reversible.

2) While the model chosen for in vivo studies is appropriate (Figures 1 and 6), the data provided require additions of further analysis of T cell activation and effector functions under the different experimental conditions used.In Figure 1 and in Figure 6, which essentially repeats the experiment in Figure 1 and tests whether a short exposure to high oxygen can rescue T cell activation under hypoxia, the authors show reduction in CD62L expression as the only parameter of T cell activation. It will be more convincing to include additional activation markers (such as CD25, CD69), especially since those are used in other experiments presented in the paper, and the sporadic usage of different markers is confusing. Assays for T cell proliferation (using Ki67, for example) and effector functions (IFNγ, TNFa, GzB levels) under hypoxia are also advisable.In the experiments for Figure 6 day 3 was chosen for normoxia; but using an earlier time point would be more congruent with the ~9 h in-vitro time window.Finally, it would be important to test if mitobiogenesis is also rescued after short oxygen exposure, so as to further identify a specific time window for mitobiogenesis essential for T cell activation.

The reviewers' comments and suggestions regarding the in vivo model are well taken. We do believe that since it is the first time that the influence of systemic low oxygen levels on T cell priming in vivo is tested, a more careful look is needed. Therefore, we invested significant resources to substantially improve the in vivo model and the subsequent phenotyping of antigen specific T cells activation status.

To better examine the priming of anti-ova CD8 T cells we modified the model from our first submission in line with a method previously used to test T cell priming in response to vaccination with lentivirus, Furmanov et al, 2010 and Furmanov et al., 2013. This model entails the injection of the OVA-lentivirus into ear pinna, intradermally. CD8 T cells activation status could then be measured by extracting cells from the deep cervical lymph nodes, which was shown to specifically drain the ear pinna. The adjusted model offered a more rapid, simple, robust method to characterize T cell activation in respect to IM immunization. Importantly the new model allows picking an experimental time frame similar to the one used in vitro.

Using this model we repeated all the in vivo experiments and completely revised both Figures 1 and 6 to include CD25, CD44, and CD62L as activation markers and Ki67/CellTrace as a marker for proliferation of the ova specific T cells. We couldn't observe CD69 elevation in our in vivo model. Most probably since CD69 elevation in T cells is a temporary event, peaks at 24 h post-activation, begin to decrease after 48h and almost entirely diminished after 96-120 h from activation. We therefore, for the sake of coherence, thought to exclude CD69 from our analysis both in vivo and in vitro. The new results are presented now in the revised Figures 1A-H, S1-C, 6K-Q, and S7A-B. Together with the experiments conducted for the first version we believe we present compelling evidence that CD8 T cell activation is compromised under systemic hypoxia.

To account for the influence of systemic exposure to low oxygen on T cell proliferation and mitochondrial-biogenesis we further developed our in vivo model. Mice were primed with LV-OVA. Three days after the viral challenge, mice were adoptively-transferred with CellTrace-labeled splenocytes from OT1/mito-Dendra2 double transgenic mice. Mice were then either kept in 8% oxygen levels for 24 hours and then transferred to atmospheric oxygen pressure for another 48 hours, or continuously kept for 72 hours in atmospheric or 8% oxygen levels. Using this approach we could directly show that low oxygen levels lead to impaired antigen-specific T cell proliferation and mitochondrial-biogenesis by analyzing CellTrace and Dendra2 intensities respectively. This key finding demonstrates that effects caused by systemic hypoxia on T cell activation are reversible and supports the notion that a short exposure to atmospheric oxygen pressure can rescue hypoxic T cells. The new results are presented now in the revised Figures 6F-J.

Numeros studies examined the effect of hypoxia on effector function of T cells. The primary focus of this paper is to understand how systemic hypoxia mechanistically affects CD8 T cell priming. We therefore thought that screening effector molecules (cytokines, grenzym), either in vitro or in vivo, will be a distraction from the primary focus of the paper. Importantly, effector molecules secretion is localized at the site of inflammation, our model was designed to examine cells in the draining lymph nodes. These CD8 T cell samples are not suitable for effector molecules phenotyping. Finally, examination of effector molecules in the small population of cells that did activate under systemic hypoxia will represent a biased phenotyping of selective cells. We therefore ask to leave effector molecules phenotyping out of the current paper.

3) In order to enhance the observations on AMPK signaling (Figure 3), it would be useful to include treatment with AMPK inhibitors in order to conclude that activated AMPK signaling is independent of the impaired activation upon respiratory restriction.

We used p-AMPK to show that the lack of ATP is not sensed differently in T-Late and T-Early and therefore is not correlated with impaired activation of T-Early. Nevertheless, following the reviewers' comment we treated T cells at different time points following activation with Compound C (also called dorsomorphin) an AMPK inhibitor. Unfortunately, naïve CD8^+^ T cells, T-Early, and T-Late cells all died following the treatment. These results indicate that both fully-activated and naïve T cells require functional AMPK to complete their activation.

4) While investigating the role of ATP transportation, it would be best to examine the activation phenotypes in ANT2-depleted or Bongkrekic acid-treated T-Early and T-Late T cells upon oligomycin treatment or hypoxia conditions Figures 3D-G).

The revised Figures 3D-G now include oligomycin treatment of ANT2KO cells.

5) The impaired cellular processes observed in T-Early, not in T-Late, T cells (Figure 4) could indicate either that mitochondrial-biogenesis is ongoing and essential for the early phase of T cell activation and is less critical in T-Late cells, or that these cellular processes in T-Late cells are independent of respiratory restriction. Data comparing these cellular processes in untreated T-Early and T-Late cells to test if mitobiogenesis has already declined in T-Late cells in comparison with T-Early cells would be useful to distinguish between these. Thus, for example, in Figure 4C, all groups could be normalised to the untreated 9 h group.Similarly, with respect to the importance of mitobiogenesis in the development of tolerance, it would be useful to examine mitobiogenesis or mitochondrial mass in T-Early and T-Late cells in Figure 4A.

Following the reviewers' comments, we added new data which include:

1) Influence of hypoxia on mitobiogenesis in T-Early: mtDendra2-derived derived CD8^+^ T cells transferred to hypoxic chamber 5h after activation demonstrated a significant reduction in Dendra2 intensity in comparison to cells (Figure 4C-D).

2) Influence of oligomycin treatment on mitobiogenesis in T-Early: at early CD8^+^ T cell activation abrogated activation (Figure 4E-F) and inhibited the increase in mitochondrial mass observed in control mtDendra2-derived CD8^+^ T cells (Figure 4G-H).

3) Influence of oligomycin treatment on mitobiogenesis in T-Late with respect to their proliferation status: we observed a substantially higher Dendra2 intensity in proliferating T-Late with respect to undivided T-Late (Figure 4I).

Overall these findings support the conclusion that respiratory restriction inhibits activation by disrupting mitochondrial-biogenesis in T-Early. Notably, we observed no reduction in dendra2 signal following oligomycin treatment in proliferating T-Late, suggesting that T-Late cells maintain mitochondrial-biogenesis during cell division independent of oxygen levels.

6) In the examination of cell size, CD25 expression and proliferation in oligomycin-treated T cells (Figure 2A), analyses of percentages of CD44- and CD25-positive cells and the expression levels of the activation markers in activated T cells would be very useful.

We revised Figures, 1I-J, S1D-E, 2B-C, 2G-H, 4A-B, 4E-F, 5A-B, and 6B-C, to include representative flow cytometry plots of FSC vs. CD25 gated on CD8^+^ T cells. We included gates that will show the % of the CD25+ cells and added bar graphs that will show a summary of the % CD25+ cells in all of the replicates.

As for the CD44 staining: since we have redone all of the in vivo experiments we included CD44 staining. Our initial in vitro analyses did not include CD44. Therefore adding CD44 as an additional activation marker will require to repeat most of our in vitro experiments. Importantly, CD44 is elevated late in the activation process in vitro and highly expressed in mature naive, memory-like cells, independent of activation signal. Given CD44 ambiguity as an activation marker we ask not to include CD44 as a marker for activation to our in vitro analyses.

7) In Figure 6, there appears to be some confusion that needs clarification. The bar graph in Figure 6F shows higher levels of effector T cells (CD62L-) in mice kept under hypoxia, which is the opposite of what the authors claim. In Figure 6G, showing CD62L expression in antigen-specific CD8 T cells, the authors claim that the data are showing reduced CD62L expression in mice exposed to high oxygen. However, the difference in the presented histograms seem to be in the number of cells acquired.

We truly apologize for this mistake. As we have redone the entire in-vivo experimentation using a new model we now present totally revised Figure 6.

8) The language in the manuscript is very difficult to follow, and revision would be very helpful. Careful editing is also needed; an example is of panels that are mentioned in the text but are missing from the figures (S3B-S3G).

The comments are well taken. We revised the manuscripts and sent it to a professional editing.

[Editors' note: further revisions were suggested prior to acceptance, as described below.]

Revisions for this paper:1) The use of oligomycin to mimic hypoxia remains a problem, since it does not simply mimic hypoxia. If it can be achieved, it is very advisable to perform these experiments with hypoxia. At the very least, the conclusions of this manuscript should be toned down substantially in the interim.

We used Oligomycin to distinguish between the indirect effects mediated by the inhibition of the electron transport chain and the TCA cycle, and the direct effect, as a result of reduced mitochondrial ATP. This cannot be achieved using hypoxia. However, we do agree with the reviewers that Oligomycin does not simply mimic hypoxia, and therefore ideally some of these experiments should be performed under hypoxic conditions. Specifically, it could be interesting to examine whether ATP-dependent matrix processes are also inhibited under hypoxic conditions. However, in practice, these experiments will require the development of complex new protocols and the acquisition of suitable devices. The major challenge in conducting these experiments under hypoxic conditions is to capture the immediate effects of hypoxia, 1-2 hours post-induction. The time taken for hypoxia to be induced varies due to multiple factors such as medium oxygen saturation, volume, chamber pressure, etc. As with other T cell signaling experiments, the result is dependent on a robust and homogeneous readout. This is hard to achieve with hypoxic chamber settings. We are currently working to develop the protocols and settings required to perform the suggested experiments and intend to explore this issue further in our follow up work.

As for our current work, we followed the reviewer request to tone down substantially the conclusions by adding the following remark in the Discussion: "Notably, some of these mechanistic observations regarding the inhibitory effect mediated by an acute respiratory restriction on CD8^+^ T cell activation were based on the application of oligomycin. Since oligomycin, only partially mimics hypoxia, follow-up work should look further into mechanistic effects induced by hypoxia induction.”

2) The cell death of early activated naive T cells induced by compound C may be a dosage-related artificial result. It is not clear whether the authors examine different doses. Some controls should be included to make the conclusion made. At the very least, the authors need to tone done their conclusions of these data in the interim.

We would like to note that AMPK signaling is not the focus of this current study. We used AMPK activation only as a marker, a proxy for the sensing of low ATP, high AMP levels in the cytosol. We note that AMPK signaling is important for T cell activation and metabolic adaptation. However, we state that under our specific experimental conditions AMPK activation levels in oligomycin-treated T-Early and T-Late cells are similar. We then claim that AMPK activation is not correlated with T-Early cells' sensitivity to respiratory restriction. Our observation that acute AMPK inhibition by Compound C treatment (we used several concentrations, 1, 5, 10, and 20 µM) leads to T cell death is in line with previous a study, Rao et al., 2016. showed that Compound C promotes, Ca^2+^ signaling-induced T cell death in an AMPK-dependent manner. Moreover, this result actually emphasizes that AMPK activation is a key component in the metabolic adaptation of T cells and highly supports a previous study from Blagih et al., 2015. We agree that our observation highlights the need to further assess the role of AMPK in T cells metabolic adaptation under chronic hypoxia conditions. To this end, we aim to generate T cell-specific AMPK knockout mice that we think will be a much more suitable tool than AMPK inhibitors.

As for our current work, we followed the reviewer request to tone down substantially the conclusions by stating in the Results section that our "results suggest that despite the important role of AMPK signaling in T cell metabolic adaptation (Blagih et al., 2015), it is not correlated with the inhibitory effects mediated by respiratory restriction in early activation".